



# A compilation of surface inherent optical properties and phytoplankton pigment concentrations from the Atlantic Meridional Transect

Thomas M. Jordan[1], Giorgio Dall'Olmo[2], Gavin Tilstone[1], Robert J. W. Brewin[3], Francesco Nencioli[4], Ruth Airs[1], Crystal S. Thomas[5], and Louise Schlüter[6]

[1]Plymouth Marine Laboratory, Plymouth, UK.
[2]Istituto Nazionale di Oceanografia e di Geofisica Sperimentale, Trieste, Italy.
[3]Centre for Geography and Environmental Science, University of Exeter, Penryn, UK.
[4]Collecte Localisation Satellites, Toulouse, France.
[5]NASA Goddard Space Flight Center, Greenbank, USA.
[6]DHI - Water and Environment, Hørsholm, Denmark.

**Correspondence:** tjor@pml.ac.uk, gdallolmo@ogs.it

**Abstract.** *In situ* measurements of particulate inherent optical properties (IOPs) – absorption ($a_p(\lambda)$), scattering ($b_p(\lambda)$) and beam attenuation ($c_p(\lambda)$) – are crucial for development of optical algorithms that retrieve biogeochemical quantities such as Chlorophyll $a$, particulate organic carbon (POC) and total suspended matter (TSM). Here we present a compilation of particulate absorption-attenuation spectrophotometric data measured underway on nine Atlantic Meridional Transect (AMT) cruises between 50° north to 50° south from 2009-2019. The compilation includes co-incident high performance liquid chromatography (HPLC) phytoplankton pigment concentrations, which are used to calibrate transects of Total Chlorophyll $a$ (Tot_Chl_a) concentrations derived from the $a_p(\lambda)$ line-height method. The IOP data are processed using a consistent methodology, and include propagated uncertainties for each IOP variable, uncertainty quantification for the Tot_Chl_a concentrations based on HPLC match-ups, application of consistent set of quality-control filters, and standardisation of output data fields and formats. The total IOP dataset consists of $\sim 310,000$ measurements at a 1 minute binning ($\sim 270,000$ hyper-spectral) and $> 700$ co-incident HPLC pigment surface samples ($\sim 600$ of which are coincident with hyper-spectral IOPs). We present geographic variation in the IOPs, HPLC phytoplankton pigments, and the $a_p$-derived Tot_Chl_a concentrations, which are shown to have uncertainties between 8-20%. Additionally, to stimulate further investigation of accessory pigment extraction from $a_p(\lambda)$, we quantify pigment correlation matrices and identify spectral characteristics of end-member $a_p(\lambda)$ spectra where accessory pigment groupings are present in higher concentrations relative to Tot_Chl_a. All data are made publicly available in SeaBASS and NetCDF formats via the following links: https://seabass.gsfc.nasa.gov/archive/PML/AMT, and https://doi.org/10.5281/zenodo.12527954.





## 1 Introduction

Inherent optical properties (IOPs) refer to the spectral absorption, scattering and attenuation characteristics of particulate and
dissolved matter in the ocean. There is a direct linkage between IOPs and the concentration and composition of optically
active marine constituents. The principal water/ocean components associated with IOPs are phytoplankton, non-algal particles
(NAP), and coloured dissolved organic matter (CDOM). These components can describe the biogeochemical processes and
rates associated with organic carbon production and export, phytoplankton dynamics in the surface and/or the upper mixed
layer of the ocean, and the responses of these over time to climatic disturbances (e.g. Werdell et al. (2018)). The spectral
signature of IOPs can also be used in algorithm development of ocean constituents from ocean colour sensors (e.g. Morel and
Prieur (1977); Tilstone et al. (2012)).

*In situ* absorption and beam attenuation spectral coefficients can be measured using ACS (hyper-spectral) or AC9 (multi-
spectral; nine channel) attenuation and absorption meters. Over recent decades, application of the underway 'flow-through'
technique (Slade et al., 2010) to ACS and AC9 meters has enabled derivation of near-continuous shipborne particulate IOP
transects for near-surface waters (Dall'Olmo et al., 2012; Boss et al., 2013; Werdell et al., 2013; Brewin et al., 2016). The
flow-through technique consists of a sequence of filtered and unfiltered/bulk measurement intervals to surface water that is
pumped into the ship. Having the filtration baseline means that dissolved and particulate IOP coefficients can be separated,
enabling derivation of particulate IOP coefficients: absorption ($a_p(\lambda)$), scattering ($b_p(\lambda)$) and beam attenuation ($c_p(\lambda)$), where
the scattering is derived from the difference between beam attenuation and absorption.

In the open ocean, Total Chlorophyll $a$ (Tot_Chl_a), a proxy for phytoplankton abundance, is the dominant control on the
spectral variability of particulate IOPs (Prieur and Sathyendranath, 1981). Ship-borne IOP flow-through data have been used
to derive near-continuous Tot_Chl_a transect data (Boss et al., 2013). This has substantially increased the spatial and temporal
coverage of data compared to discrete particulate samples (Graban et al., 2020) and, in turn, has increased the number of
Tot_Chl_a match-ups with satellite ocean colour data (Graban et al., 2020; Tilstone et al., 2021). Tot_Chl_a transect data are
calculated using the line-height of the red (676 nm) $a_p(\lambda)$ absorption peak (Boss et al., 2007), and validated or calibrated with
discrete High Performance Liquid Chromatography (HPLC) pigment concentrations before being used for satellite validation
(Brewin et al., 2016; Tilstone et al., 2021).

Beyond extraction of Tot_Chl_a concentration, the spectral features in $a_p(\lambda)$ contain information on phytoplankton ac-
cessory pigments and non-algal particles, including minerals and detritus (Bricaud et al., 2004; Devred et al., 2011; Chase
et al., 2013). Accessory pigment concentrations that can be predicted from $a_p(\lambda)$ include chlorophylls $b$ and $c$ (Tot_Chl_b and
Tot_Chl_c) as well as pigment sums of photoprotective carotenoids (PPC) and photosynthetic carotenoids (PSC) (Chase et al.,
2013, 2017; Liu et al., 2019; Teng et al., 2022; Sun et al., 2022). The increasing availability of hyper-spectral satellite ocean-
colour imagery (Werdell et al., 2019; Braga et al., 2022) provides an opportunity to improve the estimation of phytoplankton
accessory pigments from space (Dierssen et al., 2021; Kramer et al., 2022; Cetinić et al., 2024). In turn, this can improve the
estimation of phytoplankton functional types, biogeochemical cycling, and our understanding of the role of phytoplankton in
the global carbon cycle (IOCCG, 2014).





$c_p(\lambda)$ also has a range of applications in ocean-optics research. For example; $c_p(\lambda)$ provides a proxy for particulate organic carbon (POC) (Gardner et al., 2006; Rasse et al., 2017), $c_p(\lambda)$ can be used to estimate the spectral slope of the particle size distribution (Boss et al., 2001), and $c_p(\lambda)$ also provides an alternative proxy to estimate Tot_Chl_a concentration (Graban et al., 2020). Unlike $a_p(\lambda)$, which has an inverse relationship with water-leaving radiance, $c_p(\lambda)$ cannot be directly inferred from conventional space-borne ocean colour sensors. However, there is potential to retrieve $c_p(\lambda)$ more directly from the multi-angle polarimeters of the NASA Plankton, Aerosol, Cloud Ocean Ecosystem (PACE) mission (Ibrahim et al., 2012; Agagliate et al., 2023).

In this study we present a compilation of surface particulate IOPs and co-incident HPLC phytoplankton pigment concentrations from the Atlantic Meridional Transect (AMT) UK cruise program. Established in 1995, AMT has since undertaken biological, chemical and physical oceanographic research along a north-south (or occasionally south-north) transect through the centre of the Atlantic Ocean (Rees et al., 2015). Although there have been 30 AMT cruises to date, the dataset in this paper is from 2009-2019 (AMT 19, and AMT 22-29) where IOP flow-through data were collected. The AMT cruises sample a wide range of oceanographic conditions including both mid-ocean oligotrophic gyres and eutrophic shelf seas and upwelling systems (Aiken et al., 2000). Optical measurements on AMT have proved particularly valuable for satellite validation in the oligotrophic gyres, which are traditionally under-sampled (Brewin et al., 2016). AMT also facilitated the development of $a_p$-derived Tot_Chl_a transects, and related uncertainty quantification for satellite validation (Brewin et al., 2016; Tilstone et al., 2021; Graban et al., 2020). Other optical *in situ* measurements on AMT include multi-spectral $b_{bp}(\lambda)$ (Dall'Olmo et al., 2012) and above-water radiometry (remote-sensing reflectance and water-leaving radiance) (Pardo et al., 2023). Furthermore, recent AMTs (AMT 25-29) have implemented Fiducial Reference Measurement (FRM) standard above-water radiometry protocols (Lin et al., 2022).

Compilations of IOP data sets exist for both the coast (Nechad et al., 2015) and the open ocean (Barnard et al., 1998; Valente et al., 2022). However, information on the biases and uncertainties between different field campaigns, arising from differences in processing, quality control and instrumentation, is often lacking. The IOP dataset in this paper is processed using a consistent methodology refined from over 10 years development, thus reducing differences between individual cruises. The data set includes propagated uncertainties for each IOP variable, cross-validation of $a_p(\lambda)$-derived Tot_Chl_a with HPLC Tot_Chl_a, implementation of a standardised set of quality-control procedures, and standardised data release in NASA SeaBASS (a public archive of *in situ* oceanographic and atmospheric data maintained by the NASA Ocean Biology Processing Group) and NetCDF formats. In the results, we illustrate geographical variations in the IOPs, HPLC pigment concentrations, and Tot_Chl_a transects across the biogeochemical provinces that are sampled by the AMT cruises. The large number ($\sim 600$) of co-incident hyper-spectral $a_p(\lambda)$ and HPLC pigment measurements is a stand-out feature of the dataset. To this end, we then present HPLC pigment correlation matrices, which serve as a useful benchmark for accessory pigment extraction algorithms from $a_p(\lambda)$. Finally, to stimulate future algorithm application and development, we identify characteristics of $a_p(\lambda)$ spectra where accessory pigments groupings are present in higher concentrations relative to Tot_Chl_a.



## 2  Materials and Methods

### 2.1  Data coverage

The sections of AMT cruise transects where flow-through IOP data were collected are shown in Fig. 1A. The majority of the data are sampled from eight open-ocean biogeochemical Longhurst provinces, outlined in the caption. The cruises sampled from approximately 50° north to 50° south from the UK to Port Stanley (Falklands islands) or Punta Arenas (Chile). The AMT cruises were all made on UK Royal Research Ships (James Cook: AMT 19 and AMT 22, James Clark Ross: AMT 23-26 and AMT 28, Discovery: AMT 27 and AMT 29). The year of each cruise is indicated in the legend of Fig. 1A. Each cruise was typically 40 days in length, with an embarkation date between mid September-mid October. A timeline showing when flow-through IOP data were collected during each cruise, broken down by Longhurst province, is shown in Figure 1B. The times where discrete HPLC samples were taken are superimposed on the IOP data timelines.

Cruise tracks varied between different years. For example, AMT 19, 22, 24, 25 and 29 sampled the NASE and NASW provinces in the North Atlantic, whereas the other cruises sampled solely the NASE province. In the South Atlantic, AMT 24-28 sampled the SANT. Additionally, AMT 26 and AMT 27 both went further south to the island of South Georgia. AMT 25 travelled further east in the WTRA and northern part of the SATL. Due to technical issues, AMT 23 ceased IOP data collection in the WTRA, a few degrees north of the equator.

The IOP data are binned at 1-minute intervals, whereas the temporal sampling of the HPLC pigments varied between cruises, ranging from < 1 sample per day (AMT 24) to $\sim$ 3 samples per day (AMT 22 and AMT 29). The combined nine cruise dataset consists of approximately 50 weeks of data collection, incorporating > 310,000 IOP measurements ($\sim$ 270,000 hyper-spectral) with > 700 co-incident pigment samples. A cruise-by-cruise breakdown of the number of HPLC and IOP samples is given in Sect. 2.4, and a province breakdown of the number of IOP measurements is given in Sect. 3.1.

### 2.2  Measurements of flow-through particulate inherent optical properties

The absorption and attenuation spectra of seawater were measured using a continuous flow-through underway system that has been described in previous studies (Slade et al., 2010; Dall'Olmo et al., 2012; Brewin et al., 2016; Tilstone et al., 2021). The methods described in this section are based on these previous works, and summarise the steps used to measure particulate absorption ($a_p(\lambda)$), beam attenuation ($c_p(\lambda)$), and scattering ($b_p(\lambda)$) coefficients. We also present the methodology used to propagate spectrally-resolved uncertainties for each particulate IOP. The core IOP data fields, as labelled in the SeaBASS files, are summarised in Table 1.

#### 2.2.1  Instrumentation and filtration interval measurements

On all AMT cruises the optical instrumentation consisted of Wetlabs attenuation and absorption metres: either hyper-spectral ACS systems (spectral range $\sim$ 400–750 nm at $\sim$ 15 nm spectral resolution) or multi-spectral AC9 systems (9 wavelength bands at 412, 440, 488, 510, 532, 554, 650, 676, 715 nm band centres). Before being passed through the instruments, water



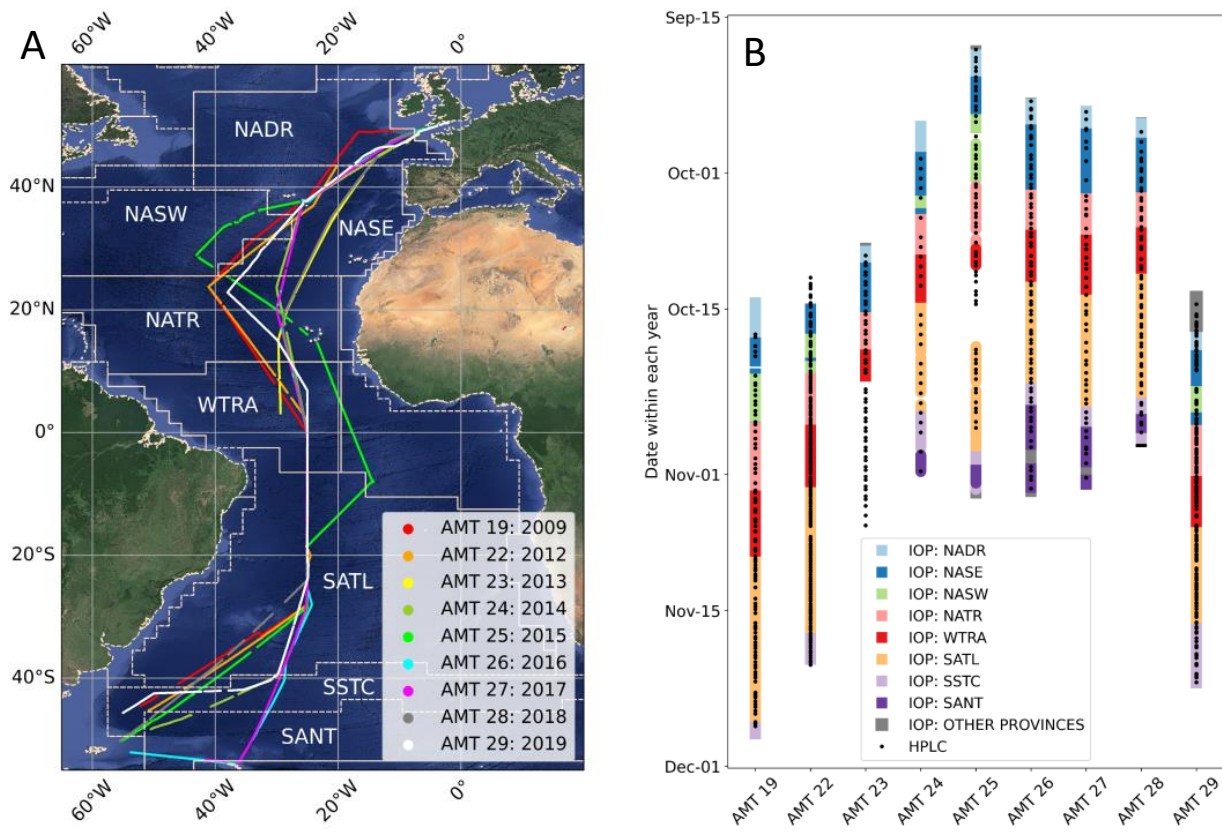

**Figure 1.** (A) Coverage map for AMT cruise transects from 2009-2019 indicating where flow-through IOP data were collected. The following Longhurst provinces, used in the data description, are indicated on the map: (NADR: North Atlantic Drift Province, NASW: North Atlantic Subtropical Gyral Province (West), NASE: North Atlantic Subtropical Gyral Province (East), NATR: North Atlantic Tropical Gyral Province, WTRA: Western Tropical Atlantic Province, SATL: South Atlantic Gyral, SSTC: South Subtropical Convergence Province, SANT: Subantarctic Water Ring Province. 'Other provinces' consists of NECS: Northeast Atlantic Shelves at the start of the cruises, and FKLD: Southwest Atlantic Shelves, ANTA: Antarctic at the end. These provinces are omitted from the majority of the data presentation due to low number of samples. (B) Timelines within each year for IOP data collected along each transect (coloured by Longhurst province) and discrete HPLC sample locations.





| Field description | Symbol/notation in paper | SeaBASS Abbreviation | Units |
|---|---|---|---|
| Particulate absorption coefficient | $a_p(\lambda)$ | ap | m$^{-1}$ |
| Particulate absorption uncertainty | $\sigma_{a_p}(\lambda)$ | ap_unc | m$^{-1}$ |
| Particulate scattering coefficient | $b'_p(\lambda)$ | bp | m$^{-1}$ |
| Particulate scattering uncertainty | $\sigma_{b'_p}(\lambda)$ | bp_unc | m$^{-1}$ |
| Particulate beam attenuation coefficient | $c_p(\lambda)$ | cp | m$^{-1}$ |
| Particulate beam attenuation uncertainty | $\sigma_{c_p}(\lambda)$ | cp_unc | m$^{-1}$ |
| Water temperature | $T$ | wt | °C |
| Salinity | $S$ | sal | PSU |
| Total Chlorophyll $a$ from particulate absorption | Tot_Chl_a: $a_p$ or $C_a(a_p)$ | Chl_lineheight | mg m$^{-3}$ |

**Table 1.** Summary of particulate IOP data fields from the ACS or AC9 instruments and core ship underway metadata. The estimation of Tot_Chl_a using the line-height method is described in Sect. 2.4, and includes calibration with HPLC pigment concentrations. The prime notation for the scattering coefficient $b'_p(\lambda)$ is used to identify that it has not been scattering- or temperature-corrected like $a_p(\lambda)$ and $c_p(\lambda)$ are. All IOP data are georeferenced and temporally binned in 1-minute intervals.

was first passed through a vortex debubbler. From AMT 26 onwards, a 45 Micro TSG Thermosalinograph (CTD) was installed to record temperature and salinity of the measurements. Prior to this, underway TSG measurements from each of the research ships were used to record temperature and salinity. A data handler (Wetlabs DH4 or DH8) was used to record the data stream, and flow-meter was used to monitor the flow-rate. The nominal depth of the water samples is assumed to be between 5-7 m across the AMT cruises.

For the first 10 minutes of each hour absorption and attenuation were measured for water that passed through a 0.2 $\mu$m filter. This was achieved by using a 3-way valve switch. For the for the rest of the hour, absorption and attenuation data were then measured unfiltered. During data processing, to avoid noisy or transient data, a minute time buffer was applied either side of the filtration interval. Median and robust-standard deviation (defined as half the difference between 84$^{th}$ and 16$^{th}$ percentiles) for attenuation and absorption were then calculated for 1-minute bins, based on a raw sample rate of 4 Hz. Sect. 2.2.3 describes the how the robust-values were used to propagate uncertainties. To provide a continuous 0.2 $\mu$m filtered baseline, filtered measurements were then linearly interpolated over the entire measurement period.

A first estimate of particulate absorption and beam attenuation was calculated by subtracting the 0.2 $\mu$m filtered measurements from the unfiltered following

$$a_{p,m}(\lambda) = a_m^B(\lambda) - a_m^F(\lambda), \qquad (1)$$
$$c_{p,m}(\lambda) = c_m^B(\lambda) - c_m^F(\lambda), \qquad (2)$$

where superscripts $B$ and $F$ denote bulk (unfiltered) sea water and filtered seawater respectively, and the subscript $m$ indicates a measured quantity prior to scattering and temperature correction.





### 2.2.2 Scattering and temperature correction

Direct measurements from the AC9 and ACS overestimate the absorption coefficient as they do not collect all of the light scattered from the incident beam. To account for scattering loss, the particulate absorption is corrected using

$$a_p(\lambda) = a_{p,m}(\lambda) - a_{p,m}(\lambda_r) \frac{b'_p(\lambda)}{b'_p(\lambda_r)}, \tag{3}$$

where

$$b'_p(\lambda) = a_{p,m}(\lambda) - c_{p,m}(\lambda), \tag{4}$$

is a first-order estimate of the particulate backscattering coefficient and $\lambda_r$ is a reference wavelength (715 nm for AC9 and 730 nm for the ACS). $b'_p(\lambda)$ is used in place of $b_p(\lambda)$ to notate the quantity as a first-order estimate (Slade et al., 2010). The reference wavelength values are chosen to extend as far as possible into the NIR as $a_p$ becomes increasing negligible with wavelength and correction via Eqn. (3) becomes increasingly accurate (Zaneveld and Pegau, 2003).

For the ACS processing, Eq. (3) is modified to account for residual temperature differences between the filtered and unfiltered
measurement interval (Slade et al., 2010). This results in a measurement equation of the form

$$a_p(\lambda) = a_{p,m}(\lambda) - \psi_T(\lambda)\Delta T - \frac{a_{p,m}(\lambda_r) - \psi_T(\lambda_r)\Delta T}{b'_p(\lambda_r)} b'_p(\lambda), \tag{5}$$

where $\psi_T$ is the temperature-dependence spectrum of pure-water absorption and $\Delta T$ is the temperature difference between filtered and unfiltered measurements (Sullivan et al., 2006). Generally, $\Delta T$ is not measured with sufficient accuracy to directly apply Eq. (5). Instead, it is derived from an optimization procedure which varies $\Delta T$ to minimize variations in $a_p(\lambda)$ at NIR
wavelengths (710-750 nm) (refer to Eq. (6) in Slade et al. (2010)). Following Slade et al. (2010), the ACS beam attenuation is also corrected for temperature dependence resulting in the measurement equation

$$c_p(\lambda) = c_{p,m}(\lambda) - \psi_T(\lambda)\Delta T, \tag{6}$$

where $\Delta T$ is predetermined from the $a_p(\lambda)$ optimization procedure.

Due to the lower spectral resolution of the AC9 (specifically, the limitation of the single wavelength band in the NIR) the
temperature-corrected measurement equations, Eqns. (5) and (6), cannot be applied. Instead, Eq. (3) and Eq. (2) are used as the measurement equations for $a_p(\lambda)$ and $c_p(\lambda)$ respectively.

### 2.2.3 Uncertainty propagation

The IOP data includes a propagated uncertainty estimate for each $a_p(\lambda)$, $b'_p(\lambda)$ and $c_p(\lambda)$ spectra, which are denoted by $\sigma_{a_p}(\lambda)$, $\sigma_{b'_p}(\lambda)$, and $\sigma_{c_p}(\lambda)$ respectively. The experimental input to the uncertainty propagation are the standard errors in the bulk and
filtered quantities: $a_m^B(\lambda)$, $a_m^F(\lambda)$, $c_m^B(\lambda)$, $c_m^F(\lambda)$ in Eqns. (1) and (2). The standard errors were calculated from the robust standard deviations of each one-minute bin, dividing by $\sqrt{N}$ where $N$ is the number of observations in each measurement bin (typically $\sim 240$). Assuming independent sources of uncertainty in the bulk and filtered time intervals, uncertainties were





propagated in quadrature for $a_{p,m}(\lambda)$ and $c_{p,m}(\lambda)$ following the standard law of uncertainty propagation (BIPM, 2008) (also applied to subsequent uncertainty propagation steps) to give uncertainties on the measured quantities, $\sigma_{a_{p,m}}(\lambda)$ and $\sigma_{c_{p,m}}(\lambda)$.

The uncertainty on the first-order approximation of the scattering coefficient, $\sigma_{b'_p(\lambda)}$, was then obtained from Eq. (4) by combining $\sigma_{a_{p,m}}(\lambda)$ and $\sigma_{c_{p,m}}(\lambda)$ in quadrature.

The propagated uncertainties for $a_p(\lambda)$ and $c_p(\lambda)$ were then derived from their respective measurement equations: Eqns. (5) and (6) for the ACS measurements and Eqns. (3) and (2) for the AC9 measurements. Specifically, for the ACS, $a_{p,m}(\lambda)$, $b'_p(\lambda)$, $\psi_T(\lambda)$ and $\Delta T$ in Eq. (5) and $c_{p,m}(\lambda)$, $\psi_T$ and $\Delta T$ in Eq. (6) were all modelled as random sources of uncorrelated

error, and propagated in quadrature to derive $\sigma_{a_p}(\lambda)$ and $\sigma_{c_p}(\lambda)$ respectively. Similarly, for the AC9, $a_{p,m}(\lambda)$ and $b'_p(\lambda)$ in Eq. (3) were modelled as random sources of uncorrelated error and propagated in quadrature to give $\sigma_{a_p}(\lambda)$. For the AC9, $\sigma_{c_p}(\lambda)$ is equivalent to $\sigma_{c_{p,m}}(\lambda)$ as $c_p(\lambda)$ is equivalent to $c_{p,m}(\lambda)$.

### 2.2.4 Quality control

To ensure high quality data, a sequence of quality-control steps were implemented to the flow-through IOP data.

1. IOP data were removed from the filtration intervals if the flow rate was < 25 (arbitrary units) or salinity was < 33 (PSU). The typical flow rates outside the filtration interval were between 30-40 units and the same flow-meter was used throughout the nine cruises, thereby ensuring consistency of the threshold. Median salinity values for each cruise transect are between 36.0 - 36.2 PSU.

2. IOP data where $a_p(420)$ was < 0 or the $a_p$-derived Tot_Chl_a concentration was < 0 were removed.

3. The $a_p$-derived Tot_Chl_a transects were then used to visually identify remaining high-frequency spikes, which likely indicate the presence of bubbles. These remaining anomalous data were then manually removed.

Typically 15-17% of measurements were removed via Step 1. This reflects the fraction of the 10 minute filtration interval within each hour of data collection, and that the flow threshold effectively separates the bulk and filtration intervals. Steps 2 and 3 only removed a small amount of additional data that were not removed via Step 1 (< 1% percent each).

## 2.3 Measurement of HPLC pigment concentrations

### 2.3.1 Overview

Discrete measurements of HPLC pigment concentrations were undertaken by four laboratories: Horn Point, University of Maryland (UMD), USA (AMT 19); NASA Goddard Space Flight Center (NASA GSFC), USA (AMT 22, AMT 29); Plymouth Marine Laboratory (PML), UK (AMT 23, AMT 24, AMT 25, AMT 26); Dansk Hydraulisk Institut (DHI) - Water and Envi-

ronment, Denmark (AMT 27, AMT 28). Following the classification in Hooker et al. (2005), the HPLC pigments correspond to primary (total chlorophylls and carotenoids) and secondary pigments (subsidiary pigments that are summed to create a primary pigment).



The primary pigments are summarised in Table 2, including their SeaBASS notation which is used throughout the paper. The commonly-used pigment sums PPC (photoprotective carotenoids) = Allo + alpha-beta-Car + Diadino + Diato + Zea

and PSC (photosynthetic carotenoids) = But-fuco + Fuco + Hex-Fuco + Perid are also provided as pigment categories. The primary pigment categories were common between laboratories. As there is some variation in the secondary pigments between laboratories and cruises, we focus on the primary pigments in the dataset presentation. Of note; the PML and DHI laboratories, following Zapata et al. (2000), do not report Monovinyl Chlorophyll *b* and Divinyl Chlorophyll *b* separately, whereas UMD and NASA laboratories do, following Van Heukelem and Thomas (2001).

| Pigment grouping | Full name of pigment | SeaBASS abbreviation |
|---|---|---|
| Chlorophylls | Total Chlorophyll *a* | Tot_Chl_a |
| | Total Chlorophyll b | Tot_Chl_b |
| | Total Chlorophyll c | Tot_Chl_c |
| Photoprotective carotenoids (PPC) | Alloxanthin | Allo |
| | Carotenes | alpha-beta-Car |
| | Diadinoxanthin | Diadino |
| | Diatoxanthin | Diato |
| | Zeaxanthin | Zea |
| Photosynthetic carotenoids (PSC) | 19'butanoyloxyfucoxanthin | But-fuco |
| | Fucoxanthin | Fuco |
| | 19'hexanoyloxyfucoxanthin | Hex-fuco |
| | Peridinin | Perid |

**Table 2.** Summary of primary phytoplankton pigments derived from HPLC included in the data set. All pigments concentrations have units mg m$^{-3}$. Total Chlorophyll *a* is the sum of Monovinyl Chlorophyll *a*, Divinyl Chlorophyll *a* and Chlorophyllide *a*. Total Chlorophyll *b* is the sum of Monovinyl Chlorophyll *b* and Divinyl Chlorophyll *b*. Total Chlorophyll *c* is the sum of Chlorophyll *c1*, Chlorophyll *c2*, and Chlorophyll *c3*.

### 2.3.2 Summary of HPLC measurement protocols

Whilst the ships were stationary, HPLC water samples were taken from surface CTD rosette 20 L Niskin bottles at mid-day stations. Additionally, between three and six samples were taken from the underway non-toxic seawater supply, collected during daylight hours from 04:00 to 22:00 local time. For both underway and Niskin samples, between 1 and 6 L of seawater were filtered onto Whatman glass fibre filters (GFF, nominal pore size of 0.7 $\mu$m), transferred to Cryovials and flash-frozen in

liquid nitrogen. On AMT 25 liquid nitrogen was not available so samples were directly placed in a freezer at -80°.

Differences in the methodologies used by the four HPLC laboratories are summarised in Table 3. Phytoplankton pigments were extracted in acetone on ice using ultra-sonification for between 3 and 20 hours. Extracts were centrifuged and/or filtered to





remove filter and cell debris. The samples were analysed using fully-automated HPLC systems with chilled autosampler (4° C)
and photodiode array detection using reversed phase C8 columns and gradient elution (Zapata et al., 2000; Van Heukelem and
Thomas, 2001). The HPLC systems were primarily calibrated using a suite of standards from DHI, Lab Products, (Denmark).
UMD and NASA also used standards from Sigma-Aldrich (USA), and UMD additionally used standards isolated in their
laboratory. NASA GSFC and Horn Point UMD included replicate HPLC measurements for $\sim 20\%$ of the stations. DHI and
PML laboratories included between 0 and 2 replicates per cruise.

| HPLC Laboratory | Horn Point UMD | NASA GSFC | PML | DHI |
|---|---|---|---|---|
| AMT cruises | AMT 19 | AMT 22, 29 | AMT 23, 24, 25, 26 | AMT 27, 28 |
| Storage temperature | -80°C | -80°C and liquid nitrogen | -80°C and liquid nitrogen | -80°C |
| Extraction solvent | 95% acetone | 95% acetone | 90% acetone | 95% acetone |
| Internal standard | Vitamin E acetate | Vitamin E acetate | Trans-ß-Apo-8'carotenal | Vitamin E acetate |
| Disruption method | Ultra-sonic probe: 12 second pulse on ice | Ultra-sonic probe: 12 second pulse on ice | Ultra-sonic probe: 35 seconds on ice | Vortex mixer and ultrasonification on ice for 10 min |
| Extraction time | 4 hours | 4 hours | 1 hours | 20 hours at 4° C |
| Clarifications | 0.45 $\mu$m Teflon syringe filter | 0.45 $\mu$m Teflon syringe filter | Centrifugation and 0.45 $\mu$m Teflon syringe filter | 0.2 $\mu$m Teflon syringe filter |
| Buffer: extract ratio | 2.5:1[1] | 3:1[1] | | 5:2[2] |
| Hardware | Agilent 1100 | Agilent 1200 | Thermo Accela | Shimadzu LC-10ADVP |
| Standard source | DHI Lab Products, Sigma, UMD | DHI Lab Products, Sigma | DHI Lab Products, Sigma | DHI Lab Products |

**Table 3.** Summary of the HPLC extraction specifications and methods used in the four laboratories. 1. 90% 28 mM TbAA, 10% MeOH. 2.
100% 28 mM TbAA. Methods are described further by Van Heukelem and Thomas (2001) (Horn Point UMD, NASA GSFC), and Zapata
et al. (2000) (PML).

## 2.4  Derivation of Total Chlorophyll *a* transects

The AMT dataset also includes near-continuous Tot_Chl_a transects derived from $a_p(\lambda)$ using a published methodology (Boss
et al., 2007; Dall'Olmo et al., 2012; Brewin et al., 2016; Graban et al., 2020). This section describes their derivation, including
calibration (de-biasing) and uncertainty quantification using HPLC Tot_Chl_a concentrations.





### 2.4.1 The line-height method

Tot_Chl_a concentrations were derived from $a_p(\lambda)$ using the line-height method (Boss et al., 2007). This method estimates
Tot_Chl_a from the phytoplankton absorption coefficient at 676 nm ($a_{ph}(676)$) as follows:

$$C_a^*(a_p) = a_{ph}(676)/0.014 = \left[ a_p(676) - \frac{39}{65}a_p(650) - \frac{26}{65}a_p(715) \right]/0.014, \qquad (7)$$

where $C_a^*(a_p)$ is an initial estimate (pre HPLC-calibration) of Tot_Chl_a derived from the ACS or AC9, $a_p(676)$ is the particulate absorption at the maximum of the peak Tot_Chl_a signal, and 0.014 $\mathrm{m^2\ mg^{-1}}$ is the chlorophyll-specific absorption coefficient. Tot_Chl_a estimates using Eq. (7) are not directly comparable between the AC9 and ACS sensors due to the nar-
rower spectral resolution of the AC9 (10 nm) with respect to the ACS (15 nm) and the residual temperature correction that is applied to the ACS (Dall'Olmo et al., 2012). Consequently, for the cruises where AC9 instruments were used (AMT 19 and AMT 28) the Tot_Chl_a values for the AC9 were adjusted to the ACS values, using the median ACS:AC9 Tot_Chl_a ratio for co-incident measurements. The ACS: AC9 median ratios were $0.69 \pm 0.08$ for AMT 19 and $0.75 \pm 0.08$ for AMT 28 where the uncertainty bounds correspond to 1 standard deviation, indicating that Tot_Chl_a derived from the AC9 is $\sim$ 25-30 % higher
than the ACS prior to the correction.

### 2.4.2 Calibration using HPLC Total Chlorophyll *a*

The calibration (de-biasing) of the $a_p$-derived Tot_Chl_a using reference HPLC concentrations follows the procedure described in Graban et al. (2020). The calibration assumes that the HPLC concentrations have higher accuracy and can be used to quantify a bias and precision uncertainty of the $a_p$-derived concentrations. Fig. 2 shows an example of the match-up procedure used
to derive the de-biased Tot_Chl_a transects from AMT 28, including the adjustment of the AC9 Tot_Chl_a values to ACS values prior to match-ups with HPLC. Prior to co-locating the ACS and HPLC data, the line-height ACS Tot_Chl_a *a* transects were median-filtered (kernel width 30 minutes) to reduce high-frequency fluctuations, typically associated with experimental noise due to bubbles in the flow-through system. The Tot_Chl_a transects were then interpolated in time coincident with the HPLC samples, enforcing ACS data to be present within a $\pm$ 15 minute time window for the match-up to be valid. Where there
were replicate HPLC samples available ($\sim$ 0 - 20% of the stations on each cruise, varying between cruises) the mean HPLC concentration was used in the match-up.

The relative residuals between line-height and HPLC Tot_Chl_a concentrations (in linear concentration space) were then calculated following

$$\rho_i = \frac{C_a^*(a_p)_i}{C_a(HPLC)_i} - 1, \qquad (8)$$

where $i$ is the sample index of the co-located data. For small $C_a^*(a_p)_i/C_a(HPLC)_i$ it can be shown that

$$\rho_i \approx \log_e[C_a^*(a_p)_i] - \log_e[C_a(HPLC)_i] \approx 2.3(\log_{10}[C_a^*(a_p)_i] - \log_{10}[C_a(HPLC)_i]), \qquad (9)$$

(i.e. the linear percentage residual is proportional to the absolute log difference). Therefore, the log-log scatter residuals in Fig. 2B are proportional the percentage residuals in Fig. 2C. The median residual ($\delta$) and robust standard deviation ($\sigma$) were then


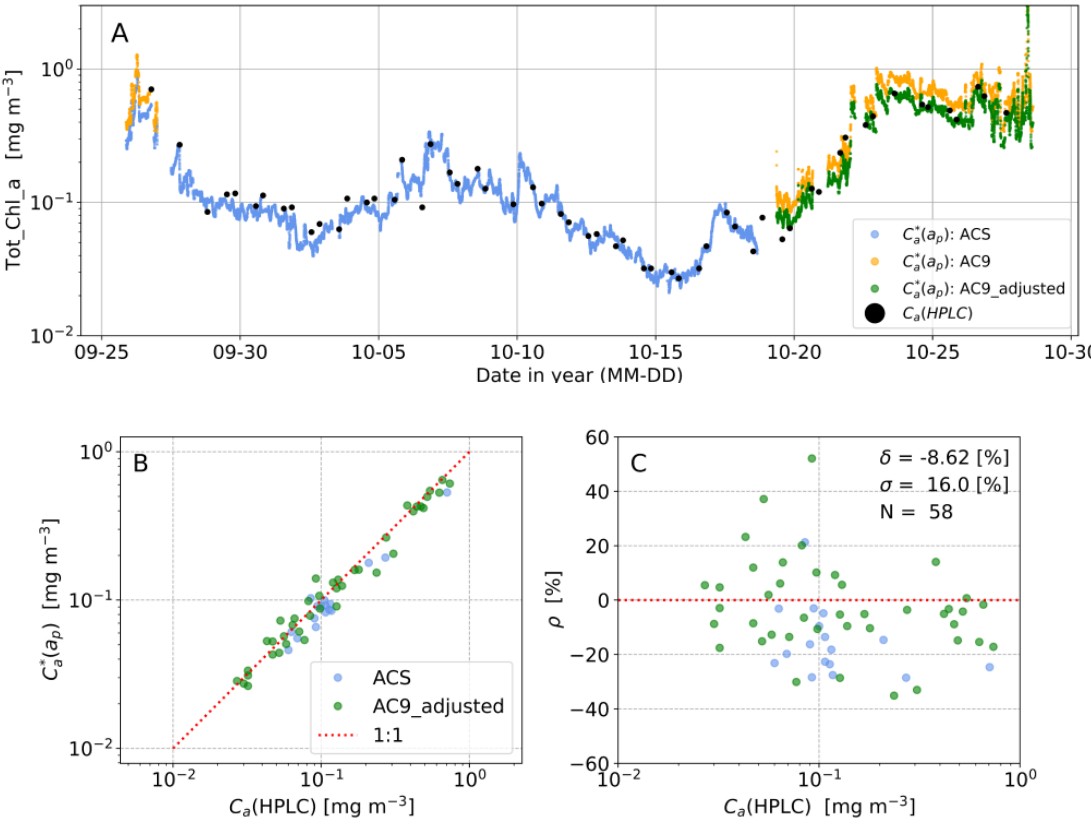

+

**Figure 2.** (A) Illustrative example of Tot_Chl_a time series from AMT 28 showing initial line-height estimates, ($C_a^*(a_p)$ median-filtered with a 30 minute kernel) from the ACS and AC9 and measurements used in the calibration. The adjusted AC9 $C_a^*(a_p)$ measurements are used where the are no ACS measurements in the final, composite, transect. (B) Corresponding log-log scatter plot showing relationship between $C_a(a_p)$ and $C_a^*(a_p)$ prior to de-biasing. (C) Residual ($\rho$) distribution in percentage units as a function of $C_a(HPLC)$, indicating median bias ($\delta$), robust-standard deviation ($\sigma$), and number of match-ups ($N$).

calculated from the distribution of $\rho$. In turn, $\delta$ was then used to adjust the $C_a^*(a_p)$ along the entire transect following

$$C_a(a_p) = (1 - \delta)C_a^*(a_p),\tag{10}$$

where $C_a(a_p)$ (no asterisk superscript) is the adjusted concentration value.

Table 4 summarises statistics for the HPLC: ACS match-ups for each individual cruise, and for the combined data set. The number of HPLC: ACS match-ups varies substantially between cruises, with AMT 19, AMT 22 and AMT 29 comprising nearly 60% of the match-up dataset. The $C_a(HPLC)$ concentration range used in the de-biasing is also given, indicating that

concentrations typically spanned two orders of magnitude for each cruise. $\delta$ ranged from -13% to 11%, with a mean value of



| Cruise | $C_a(HPLC)$ [mg m$^{-3}$] | $\delta$ [%] | $\sigma$ [%] | $\alpha$ | $\beta$ | $r^2$ | $N_{match}$ | $N_{HPLC}$ | $N_{ACS/AC9}$ |
|---|---|---|---|---|---|---|---|---|---|
| AMT 19 | 0.020 - 3.31 | -6.6 | 16.7 | 74.1 (2.4) | 0.99 (0.05) | 0.942 | 17 (72) | 91 (111) | 12,299 (31,467) |
| AMT 22 | 0.026 - 1.13 | 2.0 | 13.2 | 63.1 (2.2) | 0.98 (0.03) | 0.947 | 182 | 195 (226) | 37,009 |
| AMT 23 | 0.042 - 0.22 | -7.7 | 14.3 | 25.7 (2.9) | 0.82 (0.11) | 0.907 | 25 | 52 (53) | 11,292 |
| AMT 24 | 0.022 - 0.56 | 6.5 | 20.3 | 30.2 (4.3) | 0.86 (0.23) | 0.719 | 25 | 26 | 32,174 |
| AMT 25 | 0.029 - 3.75 | -11.8 | 9.8 | 60.3 (2.3) | 0.96 (0.04) | 0.970 | 61 | 61 (63) | 38,730 |
| AMT 26 | 0.035 - 1.49 | -13.0 | 7.6 | 69.2 (2.5) | 0.97 (0.07) | 0.907 | 77 | 80 (81) | 38,767 |
| AMT 27 | 0.029 - 4.46 | -7.4 | 11.9 | 66.1 (2.4) | 0.98 (0.06) | 0.964 | 48 | 48 | 36,482 |
| AMT 28 | 0.027 - 0.74 | -8.6 | 16.0 | 81.3 (2.4) | 1.00 (0.05) | 0.962 | 42 (16) | 58 | 24,126 (9,205) |
| AMT 29 | 0.019 - 2.58 | -4.2 | 11.1 | 85.1 (2.2) | 1.02 (0.03) | 0.964 | 148 | 148 (176) | 42,678 |
| Combined | 0.013 - 4.202 | -5.4 | 13.7 | 66.1 (2.2) | 0.98 (0.02) | 0.940 | 622 (88) | 759 (842) | 276,427 (40,309) |

**Table 4.** Summary of metrics for ACS and HPLC Tot_Chl_a match-ups from the AMT cruises. $\delta$: median percentage residual in linear concentration space. $\sigma$: robust standard deviation of residuals. $\alpha$, $\beta$, $r^2$: intercept, exponent and coefficient of determination for log-log regression of Eq. (11) with $\pm$ 95% confidence intervals for $\alpha$ and $\beta$ in brackets. $N_{match}$: number of co-located HPLC samples with ACS measurements (number of AC9 in brackets), $N_{HPLC}$: number of near-surface HPLC samples (total number including replicates in brackets), $N_{ACS/AC9}$: total number of ACS or AC9 samples (number of AC9 in brackets). The concentration range is based on the maximum and minimum $C_a(HPLC)$ value in the match-up. The bottom row presents fit metrics for all nine cruises combined. For AMT cruises where there were two ACS systems present (AMT 24 and AMT 25) the ACS Tot_Chl_a estimates were combined when matching with the HPLC data.

-5% across the nine cruises. $\sigma$ ranged from approximately 8% to 20%, with a mean value of 14%. A summary of the ACS and HPLC $C_a$ transects for each cruise are shown in Section 3.2.

The HPLC calibration implicitly assumes that there is a linear relationship between $C_a(HPLC)$ and $a_{ph}(676)$ in Eq. (7). To justify this approach, following Brewin et al. (2016), we fitted a power-law relationship of the form

$$C_a(HPLC) = \alpha a_{ph}(676)^{\beta}, \tag{11}$$

where $\alpha$ is a proportionality constant and $\beta$ a power-law exponent. $\alpha$ and $\beta$ (corresponding to the linear gradient in log-log space) were derived using Type 1 linear regression for a log10-transformation of Eq. (11). The regression fits for eight of the nine cruises showed that $\beta$ was not statistically different from unity (based on the 95% confidence interval), generally supporting use of the linear calibration. The AMT 24 cruise had a lower estimated exponent ($\beta$ = 0.82 with a confidence

interval $\pm$ 0.11), but the number of match-ups was relatively low ($N_{match}$ = 25) and the data dispersion for the calibration residuals was the highest ($\sigma \sim$ 20%) of all the AMT cruises. In addition, we performed a regression fit to Eq. (11) for the nine cruises combined, and the summary scatter plot is shown in Fig. 3A. The corresponding percentage residuals for $C_a(HPLC)$ are shown in Figure 3B, indicating that the residual distribution is broadly symmetric about zero as a function of $a_{ph}(676)$.

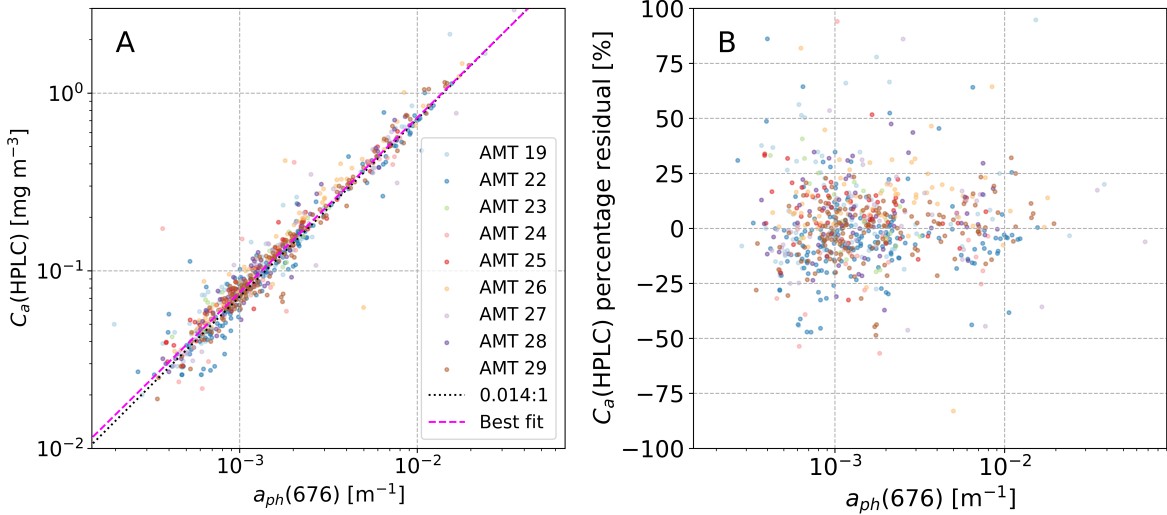

**Figure 3.** (A) Composite log-log scatter plot for $C_a(HPLC)$ against $a_{ph}(676)$, as used to derive the 'combined' power-law fit parameters $\alpha = 66.1 \pm 2.2$, $\beta = 0.98 \pm 0.02$, $r^2 = 0.94$ presented in Table 4. (B) Percentage residuals for $C_a(HPLC)$, defined as percentage difference between the observed and predicted (pink line, panel A) values. Note that, following Brewin et al. (2016), $C_a(HPLC)$ is plotted on the horizontal axis whereas in Fig. 2B it was plotted on the vertical axis.

Equivalent plots to Fig. 3A for AMT 19 and AMT 22 are shown in Brewin et al. (2016) (see their Fig. 1). Similar to Fig. 3A, 270    both the sample density and scatter-plot dispersion is lower at higher $a_{ph}(676)$ concentrations.



# 3 Results and discussion

## 3.1 Particulate inherent optical properties

Fig. 4A-C illustrate the geographical variations in hyper-spectral (ACS) IOPs, binned by Longhurst province. The absolute values of $a_p(\lambda)$ are approximately an order of magnitude smaller than $b'_p(\lambda)$ and $c_p(\lambda)$. The magnitudes of $a_p(\lambda)$, $b'_p(\lambda)$ and

$c_p(\lambda)$ all increase with Tot_Chl_a (refer to legend of Fig. 4 for province medians), and are highest in the SSTC, SANT and NADR. To compare variations in spectral shape, we integral-normalized the IOP spectra over 400-720 nm following Babin et al. (2003) and Boss et al. (2013), (Fig. 4D-F), denoted by $< a_p(\lambda) >$, $< b'_p(\lambda) >$ and $< c_p(\lambda) >$ respectively. As shown previously by Boss et al. (2001, 2013), $< c_p(\lambda) >$ approximately follows an inverse scaling relationship with the optical wavelength. $< b'_p(\lambda) >$ has a shallower spectral slope than $< c_p(\lambda) >$ due to the impact of absorption.

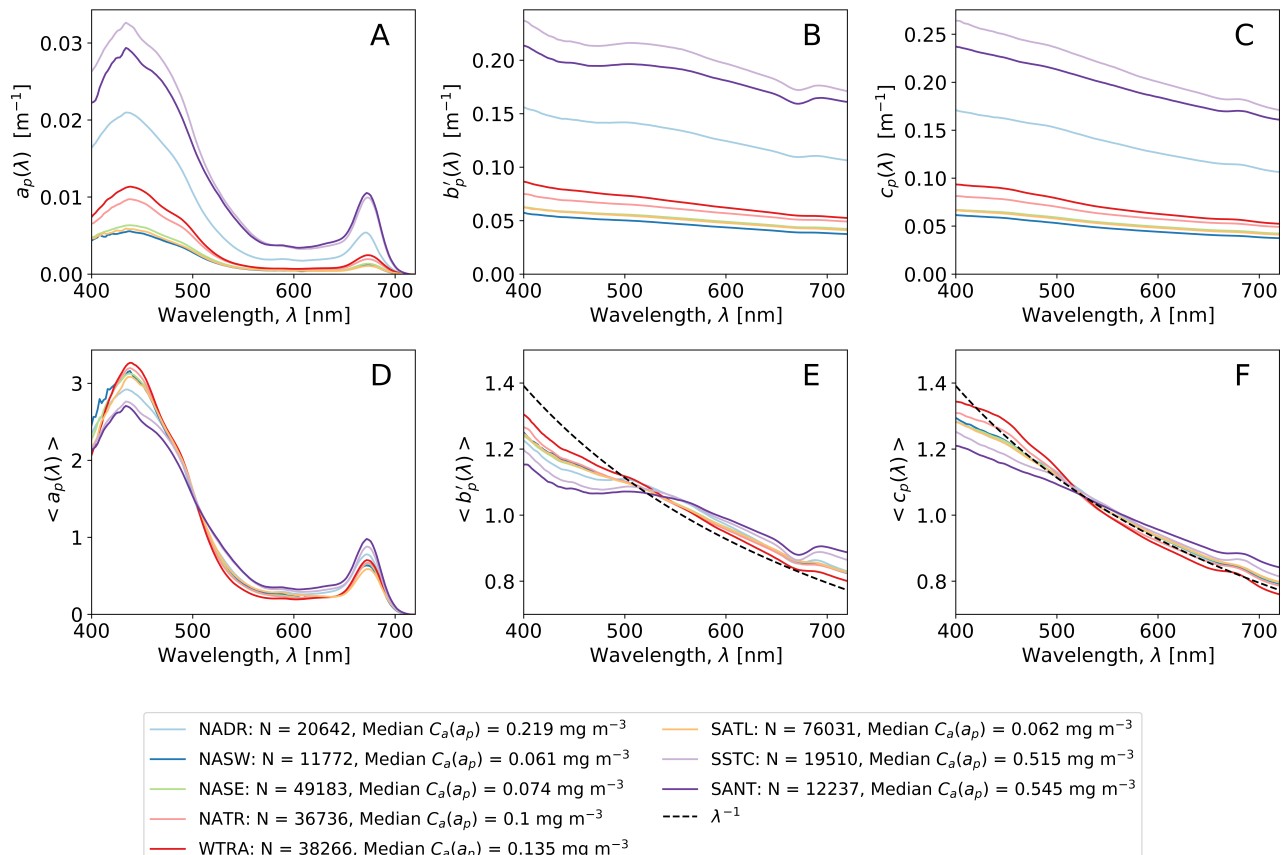

**Figure 4.** (A-C) median spectrum for particulate IOPs, binned by Longhurst province. (D-F) Median integral-normalized spectra for particulate IOPs, binned by Longhurst province. Data are combined for all nine cruises. As a reference, panels E and F include normalized inverse wavelength curves following Boss et al. (2013).





$a_p(\lambda)$ has two major characteristic peaks in the blue and red parts of the spectrum which approximately correspond to local absorption maxima of Tot_Chl_a (443 nm and 676 nm). As illustrated by Fig. 4D, the relative magnitude of the red-blue $a_p(\lambda)$ peak ratio varies with Tot_Chl_a concentration, being greater where Tot_Chl_a is higher in the NECS, SSTC, and SANT provinces. To statistically quantify this relationship, we binned the ratio $a_p(676)/a_p(443)$ (red: blue peak height) by Tot_Chl_a concentration range (Fig. 5A) and Longhurst province (Fig. 5B). In panel A, median binned values of $a_p(676)/a_p(443)$ range

from $< 0.2$ at low concentrations $(C(a_p) < 0.05$ mg m$^{-3})$ to $> 0.4$ at high concentrations $(C(a_p) > 1$ mg m$^{-3})$. In panel B, which is arranged in approximate latitudinal order, $a_p(676)/a_p(443)$ is highest at the start (NADR) and end (SSTC, SANT) of the cruises.

        The relationships in Fig. 5, occur due to presence of different phytoplankton groups, and their associated pigments (and associated absorption spectra (Bricaud et al., 2004)) being dominant at different Tot_Ch_a concentrations. In general, pico-

phytoplankton dominate at lower chlorophyll concentrations, nanophytoplankton at intermediate chlorophyll, and diatoms and dinoflagellates at higher chlorophyll (Brewin et al., 2019). Using HPLC pigment data from earlier AMTs, Aiken et al. (2009) showed that microplankton dominated the northern and southern ends of AMT, nanoplankton were dominant on the fringes of the sub-tropical gyres, with picoplankton dominant between approximately 35° north and 35° south (broadly encompassing the NASW, NASE, NATR, WTRA and SATL). In Sect. 3.3 we relate these trends to the distribution of phytoplankton pigments.

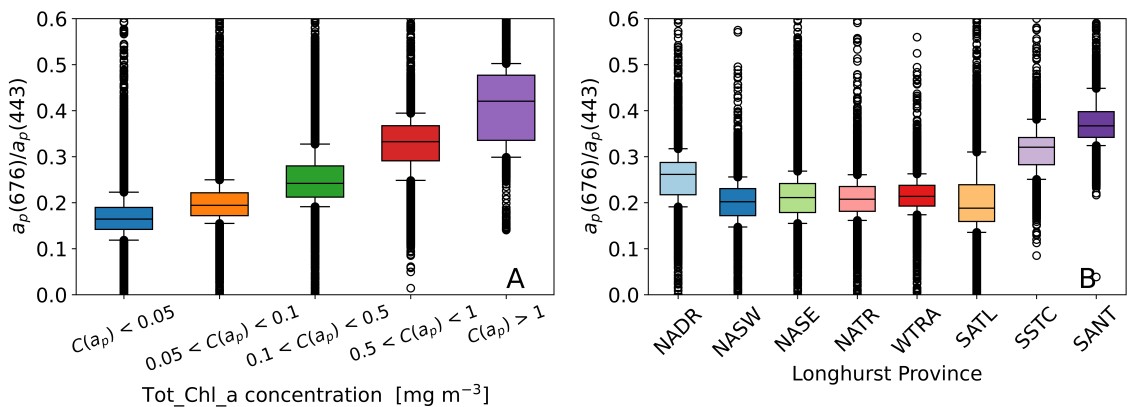

**Figure 5.** Distributions of $a_p(676)/a_p(443)$ binned by binned by Tot_Chl_a concentration (panel A) and Longhurst province (panel B). Medians: box centres, quartiles: box edges, $10^{th}$, $90^{th}$ percentiles: whisker limits, outliers: circles.

Fig. 6 shows the spectral dependence of uncertainty for $\sigma_{a_p}(\lambda)$, $\sigma_{b'_p}(\lambda)$, $\sigma_{c_p}(\lambda)$, including absolute uncertainties (median, quarterlies, $10^{th}$ and $90^{th}$ percentiles) on the top row and corresponding percentage uncertainties on the bottom row. The absolute uncertainties all have a qualitatively similar spectral shape and rapidly decrease between $\sim 400$-500 nm, are relatively flat between $\sim 500$-700 nm, and then increase again for wavelengths $> 700$ nm. The increasing uncertainties at shorter wavelengths are related to the use of tungsten-halogen bulbs in the in the ACS and AC9, which have a low signal-to-noise ratio in the blue

part of the spectrum. Absolute uncertainties are greater for $\sigma_{b'_p}(\lambda)$ as it is propagated from $\sigma_{a_p}(\lambda)$ and $\sigma_{c_p}(\lambda)$. In percentage terms, $\sigma_{a_p}(\lambda)$ is much greater than $\sigma_{b'_p}(\lambda)$ or $\sigma_{c_p}(\lambda)$ due to the lower absolute values of $a_p(\lambda)$ (Fig. 4A). For example, outside



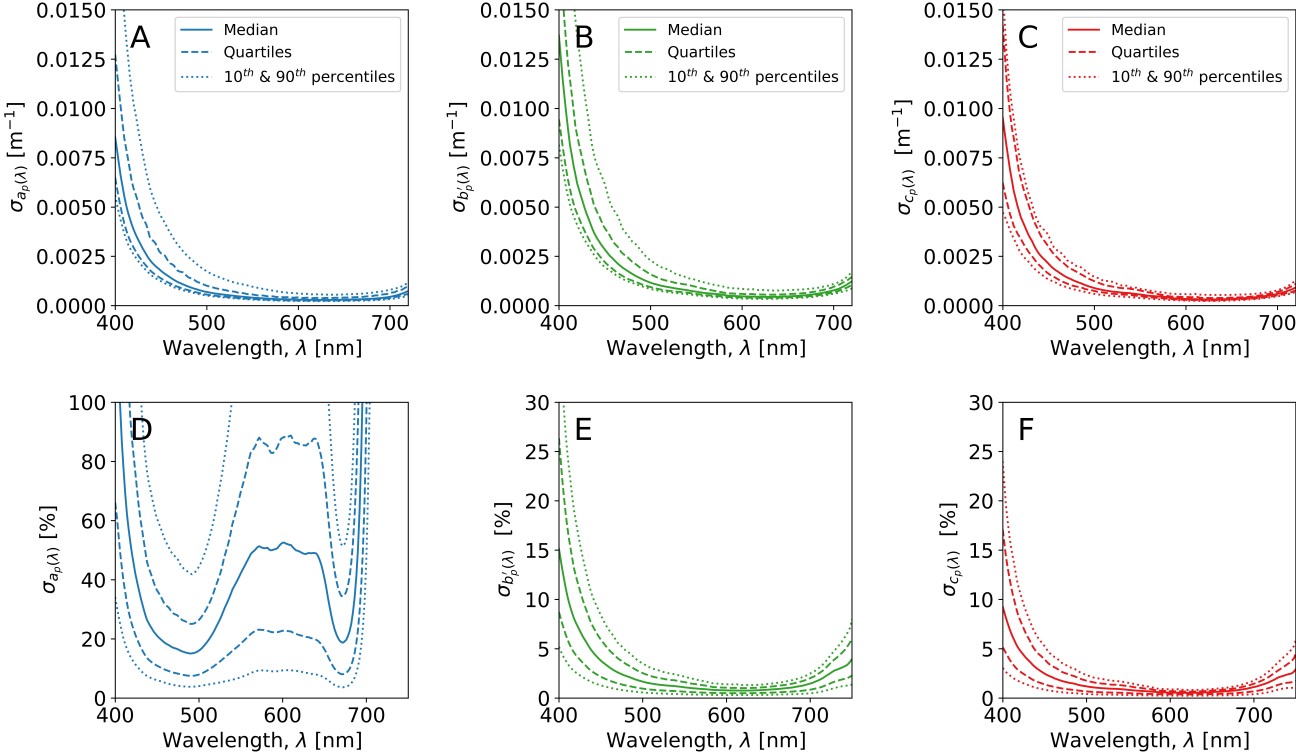

**Figure 6.** (A-C) Absolute uncertainties for particulate IOPs. (D-F) Percentage uncertainties for particulate IOPs.

of the short- and long- wavelength limits, median values of $\sigma_{a_p}(\lambda)$ typically range from 20-50%, whereas median values for $\sigma_{b_p}(\lambda)$ and $\sigma_{c_p}(\lambda)$ typically range from 2-4%.

In this section, we focused on IOP measurements from the hyper-spectral ACS system ($\sim 87\%$ of the dataset) rather than the AC9 system. Quantification of the spectral differences between ACS and AC9 systems has previously been described by Dall'Olmo et al. (2012) and Rasse et al. (2017).

### 3.2 Total Chlorophyll *a* concentrations

Fig. 7 shows the near-continuous, HPLC-calibrated Tot_Chl_a transects ($C(a_p)$) for each AMT cruise. The decreasing latitude (horizontal axis in each panel) corresponds to the north-south cruise directions, and can be referenced against the coverage map and timelines in Fig. 1. The Tot_Chl_a profiles have a broadly similar shape between cruises, with lowest Tot_Chl_a concentrations (typically between 0.01 and 0.1 mg m$^{-3}$) occurring in the northern (NASE, NASW) and southern (SATL) subtropical oligotrophic gyres, and highest values (typically order 1 mg m$^{-3}$) occurring toward the start and end of the cruises at higher latitudes in the NADR, SSTC and SANT. The SATL has the greatest variation in $C(a_p)$, with each cruise exhibiting


**Figure 7.** Latitudinal Tot_Chl_a transects for each AMT cruise with $C(a_p)$ coloured by Longhurst province. The HPLC concentrations ($C(HPLC)$) are also indicated, including the samples collected when there were no IOP match-ups. 'IOP Other' includes the data from the NECS (at the northern extremities of the cruises), and ANTA or FKLD (at the southern extremities of the cruises).





a rapid increase in concentration before entering the SSTC. Due to travelling to South Georgia, AMT 26 and AMT 27 have a
small transect section that is double-valued as a function of latitude.

The comparative differences and variability in Tot_Chl_a between AMT cruises reflects the timing, duration and location of
the cruise tracks (Fig. 1). For example, the highest Tot_Chl_a concentrations recorded in the SATL were on AMT 19, which
was later in the year, and sampled the south oligotrophic gyre for longer duration, compared to the other AMTs. AMT 28
had the fewest number of lower Tot_Chl_a values in the SATL since the cruise track sampled further west compared to the
others. AMT 27 experienced a step change in Tot_Chl_a in the SATL as the cruise track was further east and south. AMT 25
had the highest Tot_Chl_a at the start of the transect in the NADR, which had the earliest embarkation on 20 September, and
therefore captured the secondary summer bloom in the Celtic Sea. AMT 27 and AMT 28 experienced the highest Tot_Chl_a in
the SANT since these cruises went further south compared to all of the other AMTs and sampled over the sub-Antarctic front.

The HPLC Tot_Chl_a concentrations are superimposed on top of $C(a_p)$ in Fig. 7 and qualitatively illustrate the spatio-
temporal coherence between the two measurement techniques. Fig. 8 shows frequency distributions for $C(a_p)$ and $C(HPLC)$.
Both quantities have similar shape distributions, with primary peaks centred at $\sim 0.1$ mg m$^{-3}$, and secondary peaks centred
at $\sim 0.5$ mg m$^{-3}$. Cross-referencing with Fig. 7, the data comprising the primary peak primarily comes from the NASE,
NASW, NATR, WTRA, and northern part of the SATL. Similarly, data comprising the secondary peak primarily comes from
the NADR, SSTC, SANT, southern part of the SATL, and the 'Other' provinces (NECS, ANTA, FKLD) sometimes sampled
at the start and end of the cruises.

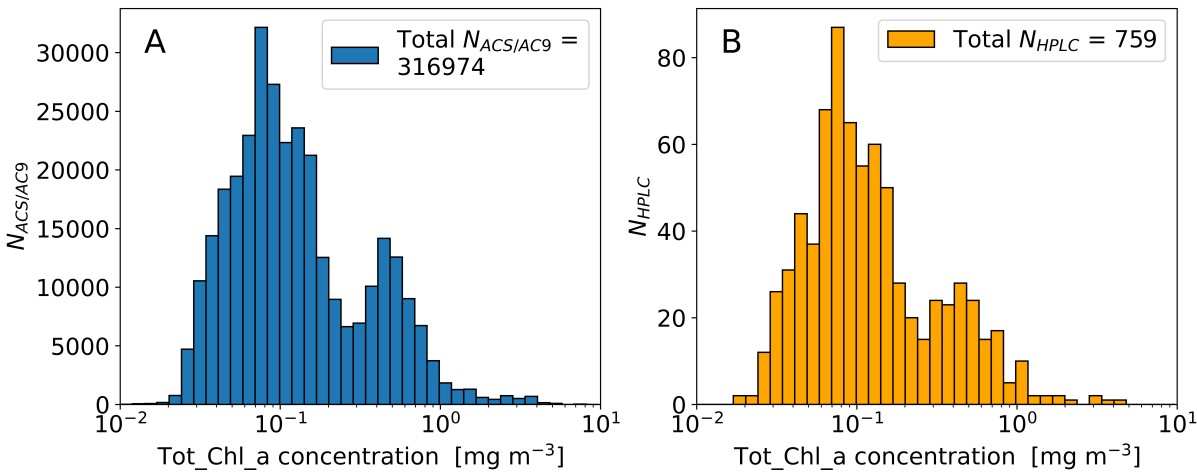

**Figure 8.** Frequency distributions of Tot_Chl_a concentration. (A) $C(a_p)$, derived from ACS and AC9 measurements. (B) $C(HPLC)$.

### 3.3    HPLC phytoplankton pigment concentrations

Fig. 9 shows the geographical distribution of phytoplankton pigment concentrations binned by Longhurst province for the
primary pigments and pigment sums (PPC and PSC) in Table 2. Fig. 10 is an equivalent plot for accessory phytoplankton





pigment concentration ratios with Tot_Chl_a. The distributions for HPLC Tot_Chl_a closely follow the trends previously shown

for the IOP transects in the previous section, with the NADR, SSTC, and SANT having highest Tot_Chl_a concentrations. The

Tot_Chl_b and Tot_Chl_c concentrations are positively correlated with Tot_Chl_a (see Sect. 3.4 for more details), and are

therefore also highest in the NADR, SSTC, and SANT provinces. In these provinces Tot_Chl_c is also higher than Tot_Chl_b,

although both are similar order of magnitude ($10^{-1}$ mg m$^{-3}$). In the other provinces Tot_Chl_b and Tot_Chl_c are both

an order of magnitude lower in concentration ($10^{-2}$ mg m$^{-3}$). The dispersion of the pigment concentration distributions is

noticeably higher in the SATL (Fig. 9F) than the other provinces. This is related to there being a rapid increase in Tot_Chl_a

concentration toward the southern end of the province (Fig. 7).

    PPC (Allo + alpha-beta-Car + Diadino + Diato + Zea) dominates over PSC (But-fuco + Fuco + Hex-Fuco + Perid) in the

lower latitude provinces (NASW, NASE, NATR, WTRA and SATL), where it is comparable in concentration to Tot_Chl_a

(order $10^{-1}$ mg m$^{-3}$). Conversely, PSC dominates over PPC in the higher latitude provinces (NADR, SSTC, SANT). This

trend was previously noted by Aiken et al. (2000) who showed explicit latitudinal dependence of the concentrations for AMT

2 and 3, and Aiken et al. (2009) who presented a compilation of pigments from AMT 1-17. Zea is the greatest contributor to

PPC in all provinces except the SSTC and the SANT, where Diadino is most abundant. Allo and Diato are generally minor

contributors to the PPC budget, with concentrations typically an order of magnitude less than the other PPC pigments. Hex-

fuco is the greatest contributor to PSC, except in the SANT where Fuco dominates the PSC budget. Broadly similar trends to

above were observed by Gibb et al. (2000) for AMT 1-5.

    A useful summary of how the phytoplankton pigments measured during AMT relate to different phytoplankton taxa and

species is provided by Aiken et al. (2009) (see their Table 1). Tot_Chl_b is present in chlorophytes, prasinophytes and

prochlorophytes, and Tot_Chl_c is present in diatoms, dinoflagellates, prymnesiophytes, chrysophytes. Zea, which dominates

the PPC budget, is generally present in cyanobacteria, shown to be *Synecococcus* or *Prochlorococcus* during previous AMT

analyses (Smyth et al., 2019). Hex-fuco, which dominates PSC budget, is generally present in prymnesiophytes. The SSTC

had the highest Hex-fuco concentrations indicative of high prymnesiophyte biomass. The SANT has high Tot_Chl_c and Fuco

indicating that diatoms dominate in this region during the AMTs.

    Previous studies have shown that photosynthetically active accessory pigment: Tot_Chl_a ratios co-vary in diatoms, prym-

nesiophytes (Hex-Fuco), cryptophytes, chlorophytes, and prasinophytes under varying growth and irradiance and nutrient (ni-

trogen) conditions (Goericke and Montoya, 1998; Schlüter et al., 2000, 2006). The high PSC: Tot_Chl_a ratios in the NADR,

SSTC and SANT (Fig. 10) suggest that the dominant phytoplankton in these provinces had the optimal photosynthetic effi-

ciency. Similarly the SSTC had the highest Hex-Fuco: Tot_Chl_a ratios indicative of prymnesiophytes and nutrient replete

conditions in this province. Other studies have shown that there can be a significant decrease in Tot_Chl_b: Tot_Chl_a ratios

due to an increase in irradiance (Ruivo et al., 2011). Since this ratio was highest in the NADR, it probably points to optimal

adaptation to the medium-range photosynthetically active radiation (PAR) experienced in this region.

    The highest Fuco: Tot_Chl_a ratios were in the SANT, which also had the greatest ratio range. Theoretically, the cell content

of Fuco in diatoms can increase with decreasing light, but the Fuco: Tot_Chl_a ratio may not be affected, due to a paralleled

increase in Tot_Chl_a (Descy et al., 2009). The large range in ratios here probably reflects the variable light and nutrient



**Figure 9.** Box plots showing distributions of phytoplankton pigment concentrations binned by Longhurst province. (Median: box centres, quartiles: box edges, $10^{th}$, $90^{th}$ percentiles: whisker limits, outliers: circles).

regimes experienced in the SANT. The NATR and WTRA had the highest PPC: Tot_Chl_a and Zea: Tot_Chl_a ratios which probably reflects the high PAR values in these provinces. Other studies have reported that PPC: Tot_Chl_a ratio increases with irradiance (Ruivo et al., 2011). Similarly, by comparison, even though the median PPC: Tot_Chl_a and Zea: Tot_Chl_a ratios were slightly lower in the NADR and SATL, these ratios exhibited the greatest spread in values. Some studies also report that Zea: Tot_Chl_a ratios increase with increasing irradiance and/or change significantly under fluctuating irradiance (Descy et al., 2009), which may reflect the high range in the Zea: Tot_Chl_a ratios in the NADR and SATL.



**Figure 10.** Box plots showing distributions of phytoplankton accessory pigment concentration ratios with Tot_Chl_a binned by Longhurst province. (Median: box centres, quartiles: box edges, $10^{th}$, $90^{th}$ percentiles: whisker limits, outliers: circles).

## 3.4  Pigment correlation structure

Accessory pigment concentrations can be estimated either from empirical relationships with $a_p(\lambda)$ (or decomposition or derivatives of $a_p(\lambda)$) or from Tot_Chl_a (Chase et al., 2013, 2017; Liu et al., 2019; Teng et al., 2022; Sun et al., 2022). The correlation structure between HPLC Tot_Chl_a and accessory pigments measured on AMT is therefore a useful benchmark for measuring the performance of $a_p(\lambda)$-related accessory pigment algorithms that may be applied to the data set, as well as constraining how





accurately accessory pigments can be predicted from Tot_Chl_a alone. Additionally, the correlation structure between phytoplankton pigments can be used to derive phytoplankton community composition (Kramer and Siegel, 2019; Kramer et al., 2020). Towards these objectives, Fig. 11A shows the Spearman correlation matrix ($r$) between primary pigment concentrations and their concentrations ratios relative to Tot_Chl_a, and Fig. 11B shows the corresponding coefficient of determination matrix ($r^2$). Both $r$ and $r^2$ are shown, as the former indicates positive or negative correlations, and the latter indicates the predictive

strength of the relationship (i.e. the proportion of variance in the dependent pigment concentration or concentration ratio that can be explained by the known pigment). Correlations for the pigment sums, PPC and PSC, are also shown as they can indicate pigment packaging effects, as well as species composition and adaptation to light conditions (Eisner et al., 2003). We note that the correlation strength between PPC and PSC and their constituent pigments is increased due to the pigment sums being partially determined by the respective accessory pigment concentration.

The accessory pigment concentrations are all positively correlated with Tot_Chl_a (top row in Fig. 11A), and, to lesser degree, with each other (upper right triangle in Fig. 11A). Tot_Chl_c and PSC have the strongest correlations with Tot_Chl_a ($r^2$ = 0.90 and 0.91 respectively), whereas Tot_Chl_b and PPC have weaker correlations ($r^2$ = 0.46 and 0.59 respectively). Tot_Chl_c also has stronger correlations with other pigments than Tot_Chl_b. The strength of the Tot_Chl_b correlations is believed to be partially reduced by a systematic difference in Tot_Chl_b concentrations that were noted between HPLC

laboratories. Specifically, at low-medium values of Tot_Chl_a (0.05 mg m$^{-3}$ or less), PML Tot_Chl_b concentrations are systematically higher than the other laboratories, with median Tot_Chl_b concentrations of 0.019 mg m$^{-3}$ for PML and 0.008 mg m$^{-3}$ for DHI and NASA combined. We therefore recommended evaluating Tot_Chl_b relationships on a laboratory-by-laboratory basis in future work. Out of the individual carotenoids, Allo, alpha-beta-Car, Diadino, Hex-Fuco and Fuco all have $r^2 > 0.4$ for their correlation strength with Tot_Chl_a.

The main point of showing the correlation matrix for absolute values of the pigment data set (upper diagonal in Fig. 11A and B) is that there can be high correlations between key phytoplankton group specific pigment markers and many other accessory pigments. In turn, this illustrates the challenge for forward or inverse modelling of optical data to retrieve phytoplankton groups especially with absorption peaks or reflectance troughs at neighbouring bands, as there may be multiple model combinations for similar species Moisan et al. (2011). The problem becomes more tractable to solve when the pigment data are normalized

to Tot_Chl_a (lower diagonal in Fig. 11 A and B), as there are fewer significant correlations with Tot_Chl_a, and of those that are significant, some have negative correlations (e.g. Zea, alpha-beta-Car, PPC), whereas others are positive (Fuco, Tot_Chl_c, PSC).

The elements in the left column differ from the rest of the lower left triangle matrix as they show correlations between absolute Tot_Chl_a and accessory pigments normalized by Tot_Chl_a. These correlations are lower than the corresponding

normalized concentrations in the top row, with $r^2 < 0.4$ in all cases. The normalized Zea concentration is strongly determined by the normalized PPC concentration ($r^2$ = 0.97), which is related to how it dominates the PPC budget across the majority of the provinces (Fig. 9 and 10). To a lesser extent, the normalized Hex-fuco concentration is determined by the normalized PSC concentration ($r^2$ = 0.69).



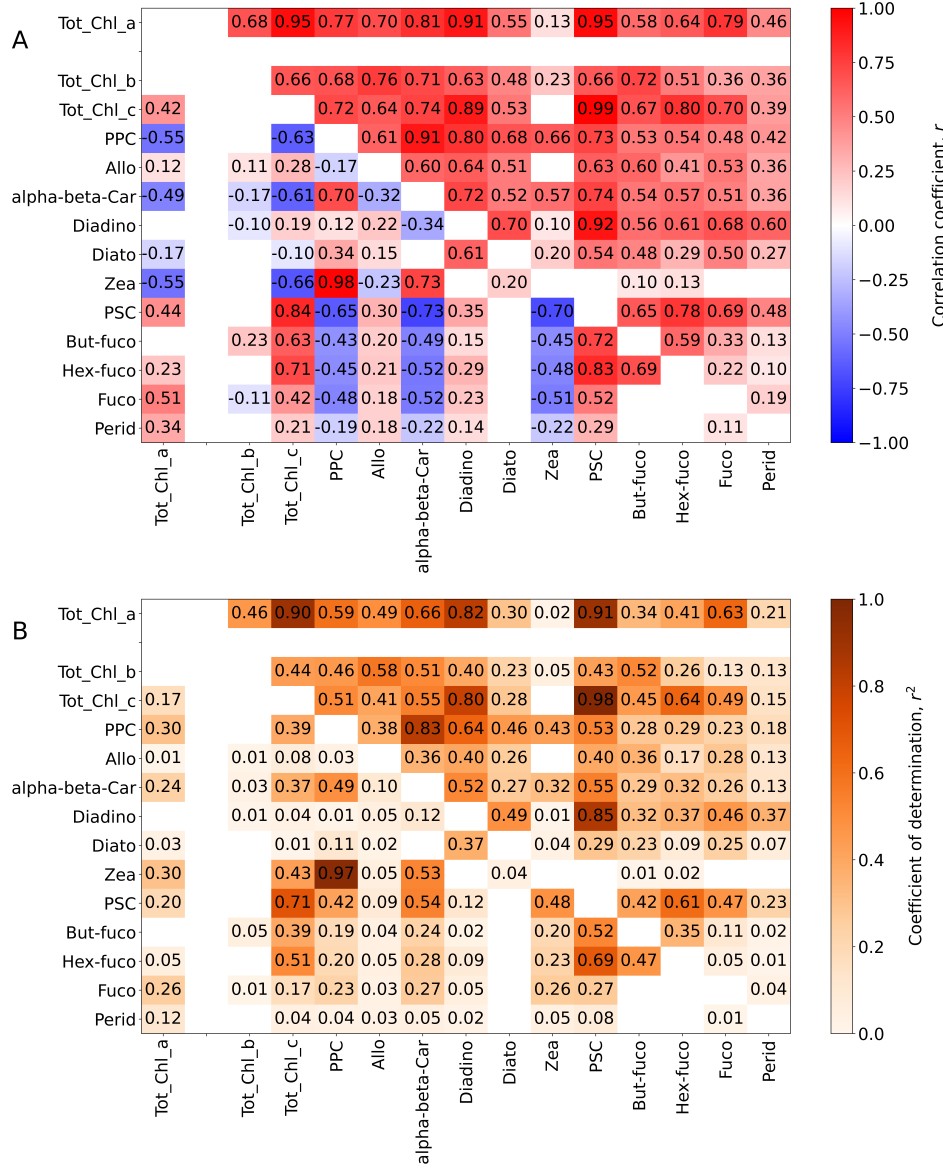

**Figure 11.** (A) Spearman correlation ($r$) matrix for phytoplankton pigment concentrations/concentration ratios relative to Tot_Chl_a. (B) Coefficient of determination ($r^2$) matrix for phytoplankton pigment concentrations/concentration ratios relative to Tot_Chl_a. Elements above the central diagonal are correlations between absolute pigment concentrations (linear concentration units). Elements below the central diagonal are correlations between pigment concentrations normalized by Tot_Chl_a, with the exception of the left column (Tot_Chl_a correlation elements), where normalization is only applied to the accessory pigments. For clarity of presentation, the Tot_Chl_a elements (top row and left column) are separated from the rest of the matrix. The number of samples was $N_{HPLC} = 759$. Correlation elements which had $p$ values > 0.01 (assessed using a two-sided test with null hypothesis that the slope is zero) were deemed statistically insignificant and were not included in the figure. The presentation broadly follows Kramer and Siegel (2019).




Similar correlation matrices assessed on different HPLC datasets are provided in Kramer and Siegel (2019) and Teng et al.
(2022) (Global datasets) and Sun et al. (2022) (data from Bohai Sea, Yellow Sea, and East China Sea). Overall, we observe
similar structure to these studies with absolute pigment concentrations having generally positive correlations, and normalized
pigment concentrations (presented only by Kramer and Siegel (2019)) having a mix of positive and negative relationships, as
well as weaker correlation strengths. Given the pigment diversity across the Longhurst provinces (Figs. 9 and 10), subsequent
correlation analysis of the AMT pigment dataset may consider binning by geographical region and/or Tot_Chl_a concentration
range.

## 3.5 Spectral end-members in $a_p(\lambda)$: accessory pigment match-ups

Using the methodology described in Sect. 2.4, there are $\sim 600$ hyper-spectral match-ups between $a_p(\lambda)$ and HPLC samples
across the nine cruises. This co-located IOP: HPLC dataset is a valuable resource to test existing phytoplankton group or
functional type algorithms (e.g. Chase et al. (2013); Teng et al. (2022); Sun et al. (2022)), or develop new algorithms to
extract accessory pigment concentrations from hyper-spectral $a_p(\lambda)$. To motivate this objective, we now illustrate how $a_p(\lambda)$
and $< a_p(\lambda) >$ can vary for data subsets with high concentration ratios of Tot_Chl_b, Tot_Chl_c, PPC and PSC relative to
Tot_Chl_a. An overall goal is to relate to the choice of wavelengths that have previously been used to estimate accessory
pigments (Chase et al., 2013, 2017), which, in turn, were based on absorption spectra for the individual pigments (Bricaud
et al., 2004). Additionally, the $a_p(\lambda)$ and $< a_p(\lambda) >$ with high of accessory-pigment concentration ratios will serve as a useful
baseline for determining phytoplankton community spectra from hyper-spectral ocean colour missions such as PACE (Cetinić
et al., 2024), the Geosynchronous Littoral Imaging and Monitoring Radiometer (GLIMR) mission (Dierssen et al., 2023) and
the Surface Biology and Geology (SBG) (McClain et al., 2022).

We identified $a_p(\lambda)$ and $< a_p(\lambda) >$ associated with high concentration ratios of accessory pigments as the subset of spectra
which exceed the $90^{th}$ concentration-ratio percentile ($P_{90}$) of the HPLC concentration values. Median $a_p(\lambda)$ and $< a_p(\lambda) >$
were then calculated across the subset of spectra that exceeded the $P_{90}$ thresholds and are shown in Fig. 12A and B respectively.
The global median spectra, $a_p(\lambda)$ and $< a_p(\lambda) >$, are also shown as a reference in Fig. 12A and B, and the median $< a_p(\lambda) >$
spectra was used to calculate the residuals in Fig. 12C. We note that, in this context, the 'global' median is a statistical average
of the shape of $a_p(\lambda)$ and $< a_p(\lambda) >$ and does not correspond to median pigment concentrations. It is also typical of small
(pico-) phytoplankton (Ciotti et al., 2002) encountered during the AMT cruises.

Fig. 12A shows that the magnitude of $a_p(\lambda)$ is greater for the high Tot_Chl_c and high PSC data subsets. This observation
can be related to geographic distributions of the concentration ratios relative to Tot_Chl_a (Fig. 10) which show that high
Tot_Chl_c spectra are most likely to be from the NADR, SSTC and SANT, and high PSC spectra are most likely to be from the
SANT or SSTC (i.e. the more productive regions). On the other hand, the high PPC spectra, which have the lowest magnitude
for the median spectrum in Fig. 12A, are most likely to be from the NATR which is oligotrophic and dominated by pico-
phytoplankton or the WTRA which oscillates between periods of high productive equatorial upwelling and low production and
is often dominated by nano-phytoplankton (Barlow et al., 2023). The similarity between high Tot_Chl_c and high PSC spectra
can be explained by their high positive correlations in Fig. 11 ($r^2 = 0.71$ for the Tot_Chl_a-normalized concentrations). The

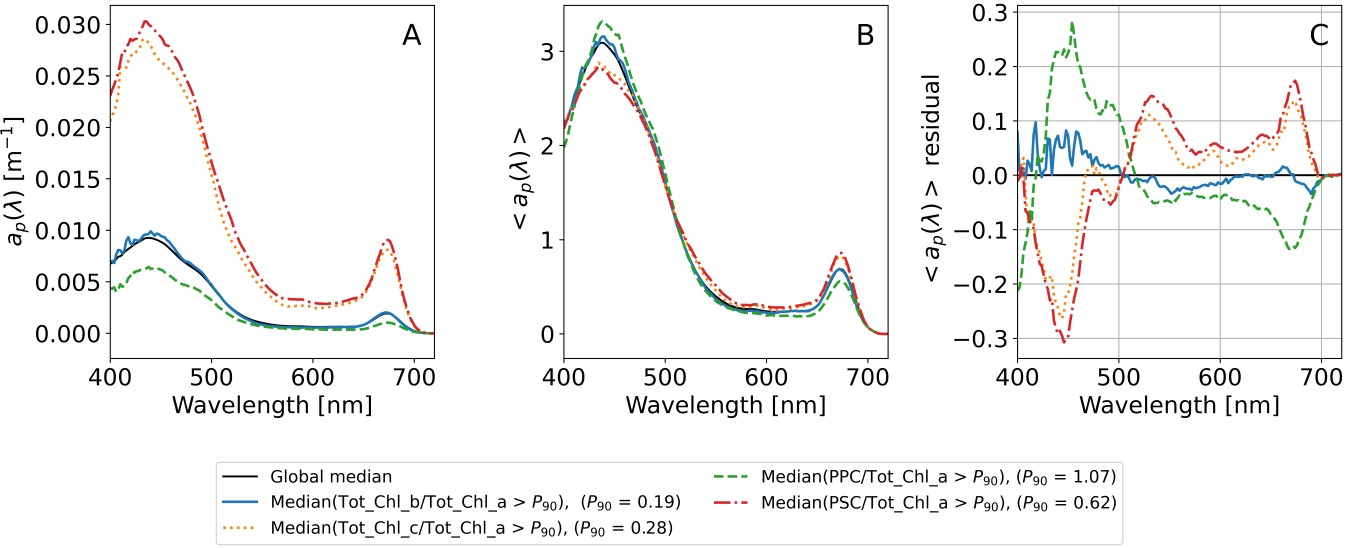

**Figure 12.** Median particulate absorption spectra and residuals for the top 10 percentiles of accessory-pigment (Tot_Chl_b, Tot_Chl_c, PPC and PSC) to Tot_Chl_a concentration ratios. (A) $a_p(\lambda)$. (B) $< a_p(\lambda) >$. (C) Residual of $< a_p(\lambda) >$ from the global medians. $P_{90}$ thresholds for accessory-pigment Tot_Chl_a ratios are provided in the legend. 'Global median' refers to an averaged property of the dataset, which is all sampled in the Atlantic Ocean.

median high Tot_Chl_b spectrum is close to the global median, and consistent with the lack of clear geographical trend in Fig. 10. The spectral shape of the high Tot_Chl_c and high PSC (best illustrated by the $< a_p(\lambda) >$ median spectrum in Fig. 12B) have higher red-blue peak ratios than high Tot_Chl_b and high PSC spectra.

The study by Chase et al. (2013) decomposed $a_p(\lambda)$ spectra into component Gaussian functions, which physically relate to absorption peaks of individual phytoplankton pigments (Bricaud et al., 2004). In their study, this was formulated as an optimization, where peak wavelength sensitives for correlations between $a_p(\lambda)$ and respective Tot_Chl_b, Tot_Chl_c, PPC and PSC concentrations were derived. Overall, there is some correspondence between optimized $a_p(\lambda)$-Gaussian peaks in Chase et al. (2013) and the $< a_p(\lambda) >$ residual peaks in Fig. 12C. For example, the single optimized peak locations for PPC and PSC in Chase et al. (2013) are at 492 nm and 523 nm respectively, which are both close to local maxima for PPC and PSC in Fig. 12C. Their optimized peak locations for Tot_Chl_c (585 nm and 639 nm) are also close to (minor) local maxima in Fig. 12C. The correspondence is less clear for Tot_Chl_b (due to the similarity with the global median spectrum), although we note that there is a local maximum in Fig. 12C close to the optimized peak of 661 nm in Chase et al. (2013). The subtle differences between our study and theirs, likely reflects the conditions in the Atlantic Ocean during boreal autumn and austral spring compared to snapshots of more variable conditions from a geographically global database (Boss et al., 2013). The data available from our AMT dataset will help to refine regional to basin scale to global algorithms for the decomposition of major phytoplankton groups from $a_p(\lambda)$.



## 4 Summary

In this study, we presented a decadal (2009-2019) dataset of particulate IOPs and HPLC phytoplankton pigments from the surface waters of the Atlantic Ocean, sampled during nine AMT cruises from 50° north to south. The dataset was processed in a consistent manner and included propagated uncertainties for each particulate IOP. We presented geographical variations in the IOPs and phytoplankton pigments, that builds on previous research of these data from historic AMT cruises (Aiken et al., 2000; Gibb et al., 2000; Aiken et al., 2009; Dall'Olmo et al., 2012; Brewin et al., 2016; Graban et al., 2020; Tilstone et al.,

2021). The dataset also included near-continuous Tot_Chl_a transects derived from particulate absorption which were shown to have uncertainties between 8-20%.

The majority of the IOP measurements (> 270,000) were collected with hyper-spectral ACS systems and therefore have particular significance for biogeochemical algorithm development in the era of hyper-spectral ocean colour remote sensing. To this end, we presented analysis of pigment correlation structure (which serves a useful benchmark for deriving accessory pigments

from particulate absorption), demonstrating that correlation strengths are significantly reduced if pigment concentrations are normalized by Tot_Chl_a. Finally, to stimulate future investigations, we illustrated variation in particulate absorption spectra for high concentrations of accessory pigments relative to Tot_Chl_a. This compiled dataset will be useful to a range of research communities, including remote-sensing scientists, marine ecologists, biogeochemical modellers and climate scientists alike.

## 5 Code and data availability

IOP processing code for each cruise, including match-ups with HPLC pigments, are provided in the following public github repositories: AMT_19_code, AMT_22_code, AMT_23_code, AMT_24_code, AMT_25_code, AMT_26_code, AMT_27_code, AMT_28_code, AMT_29_code. The core IOP processing code is written in Octave and consists of three steps. Step 1 consists of calibration from Level 0 (raw) to Level 1 data (SI units). It also includes data binning in 1-minute intervals, and then sorting the data into daily files. Step 2 consists of the ACS and AC9 physical data processing steps outlined in Sect. 2.2 and also

includes initial estimation of Tot_Chl_a using the line-height method and extraction of ship underway metadata. Step 3 merges daily data files from Step 2 into a single data file for each cruise, and includes adjustment of AC9 Tot_Chl_a to ACS Tot_Chl_a concentrations. In Step 4, file conversion to NetCDF is performed in Python. In Step 5, provided as a Jupyter notebook, match-ups and calibrations with HPLC pigment data are performed, as described in Sect. 2.4. Step 5 also includes translation of DHI and PML pigment naming conventions to NASA SeaBASS standards, incorporation of HPLC metadata in the NetCDF files,

and cruise-specific filtering of pigment data (e.g. in some cases, removal of HPLC samples in deeper waters or measured with different filter sizes).

The data described in this manuscript are available in two separate repositories. The SeaBASS formatted data are available at https://seabass.gsfc.nasa.gov/archive/PML/AMT under DOI: 10.5067/SeaBASS/AMT/DATA001 and citation guidance is provided on the SeaBASS website at https://seabass.gsfc.nasa.gov/wiki/Access$_Policy$. NetCDF formatted data are available at

https://doi.org/10.5281/zenodo.12527954 (Zenodo) and can be cited under Jordan et al. (2024). The SeaBASS data for each cruise consists of separate files for each ACS or AC9 system and the HPLC pigments. There is a single NetCDF file for each

cruise which also contains extended ship underway metadata. Code for the data analysis and plot functions are in the following repository: AMT_IOP_plots, which also contains Jupyter notebook showing how to access the data fields in the SeaBASS and NetCDF files.

*Author contributions.* TMJ: Data curation, Formal analysis, Investigation, Methodology, Software, Validation, Visualization, Writing – original draft. GDO: Conceptualization, Data curation, Formal analysis, Funding acquisition, Investigation, Methodology, Software, Supervision, Validation, Visualization, Writing – review and editing. GT: Data curation, Funding acquisition, Investigation, Methodology, Supervision, Writing – original draft. RJWB: Data curation, Funding acquisition, Investigation, Methodology, Writing – review and editing. FN: Data curation, Investigation, Software. RA: Formal analysis, Writing – review and editing. CT: Formal analysis, Writing – review and editing. LS:
Formal analysis, Writing – review and editing.

*Competing interests.* The authors declare no competing interests.

*Acknowledgements.* The authors would like to thank the captain and crews of the UK Royal Research ships which were used for data collection. This research was part of the AMT programme which is funded by the UK Natural Environment Research Council through its National Capability Long-term Single Centre Science Programme, Climate Linked Atlantic Sector Science (grant number NE/R015953/1).
Additional funding was provided by the following European space agency contracts: AMT4SentinelFRM (ESRIN/RFQ/3-14457/16/I-BG), AMT4OceanSatFlux (4000125730/18/NL/FF/gp), and AMT4CO2Flux (4000136286/21/NL/FF/ab). This is contribution number 408 of the AMT programme.



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
