# Peer review of "A compilation of surface inherent optical properties and phytoplankton pigment concentrations from the Atlantic Meridional Transect"

_Earth System Science Data, 2024_

## Referee Comment (RC2)

Piotr Kowalczuk
Institute of Oceanology
Polish Academy of Sciences
Ul. Powstańców Warszawy 55
PL – 81 - 712 Sopot

October 25, 2024

Dr Annick Bricaud
Earth System Science Data
Editorial Support Team

Attn: Review of the manuscript by Thomas M. Jordan, Giorgio Dall'Olmo, Gavin Tilstone, Robert J. W. Brewin, Francesco Nencioli, Ruth Airs, Crystal S. Thomas, and Louise Schlüter entitled "A compilation of surface inherent optical properties and phytoplankton pigment concentrations from the Atlantic Meridional Transect" submitted to Earth System Science Data and coded essd-2024-267.

Dear Dr Bricaud,

After reading the manuscript by Jordan et al., submitted to Earth System Science Data and coded essd-2024-267 **I recommend to consider this paper for publication in this journal almost as is.**

General opinion

Authors have presented and described in thorough details an impressive data set consisting with more than 300 000 measurements points of spectral values of particulate absorption $a_p(\lambda)$, scattering $b_p(\lambda)$ and beam attenuation $c_p(\lambda)$ coefficient, measured along track of nine Atlantic Meridional Transect cruises between 2009-2019. Measurements were conducted with use spectral absorption and attenuation meter ac-9 at nine spectral channels (SeaBird Inc. USA) or its hyperspectral version asc, in the spectral range ~ 400–750 nm at ~ 4 nm spectral resolution. Presented data set includes ca. 700 coincident measurements of phytoplankton pigments concentrations with use of the high performance liquid chromatography methods (HPLC). Phytoplankton pigments concentrations measured in collected water sample were used to derive continues transect of the total chlorophyll-a concentration in the surface water based on regression between HPLC total chlorophyll-a concentration and absorption line height at 676 nm parameter calculated from red part of particulate absorption spectrum. This study covered very detailed spectral uncertainty budget of optical measurements and thorough discussion of uncertainty associated with inter laboratory results of the HPLC analysis. Author also included the correlation matrix between total chlorophyll-a concentration and identified accessory pigments and identified spectral characteristics of end-member $a_p(\lambda)$ spectra where accessory pigment groupings were present in higher concentrations relative to Tot_Chl_a. Authors have also presented distribution of selected values of bio-optical parameters in sampled biogeographic provinces of the Atlantic Ocean which make this study not only presentation of data collection but additionally very detailed research paper presenting very valuable bio-optical characterization of the Atlantic Ocean between 50 deg. N and 50 deg. S.

I am impressed by the quality of the data set, and how authors have presented it. I am truly confident that this study shall be published as is. There are very few minor mistakes that I have spotted, by those can be corrected during proof edition. Manuscript does not need any further review.

Congratulations to Authors. Very well done.

Detailed comments

I have spotted just two minor mistake:

Page 4, line 114

Is: "ACS systems (spectral range ∼ 400–750 nm at ∼ 15 nm spectral resolution)…"

Actually the spectral resolution of the acs instrument is ca. 4.3 nm. According to technical specification provided by manufacturer acs measures absorption and attenuation in the spectral range 400–750 nm at 80 spectral channels which makes 4.3 nm spectral spacing between channels. Please correct.

Page 7, Equation 4

Is: $b'_p(\lambda) = a_{p,m}(\lambda) - c_{p,m}(\lambda)$,

Shall be

$b'_p(\lambda) = c_{p,m}(\lambda) - a_{p,m}(\lambda)$

Use of Equation 4 as written in the original manuscript would results in negative values of spectral scattering coefficient, as spectral attenuation coefficient values are greater than spectral absorption coefficient values. Please correct.

Best regards

Piotr Kowalczuk

---

## Author Comment (AC1)

**Author response to essd-2024-267**

Dear ESSD editors,

We would like to thank our reviewers, Emmanual Boss and Piotr Kowalczuk, for their constructive and critical feedback. We have addressed their comments to the best of our ability and believe that our MS is significantly improved from the previous submission.

Our responses to their specific comments are in blue. We will also attach a marked-up version of our MS, indicating where changes have been made.

Best regards,

Tom Jordan (on behalf of all authors).

**Review 1**

L72 There are several cruise specific or program specific compilations. e.g. CoastLOOK has been recently published in this journal.

We have added the reference for CoastLook (Massicotte et al. 2023). We have also added 'e.g.,' in the reference parenthesis to make it clear that other references also exist.

L117 Is the temperature measured at the intake or within the flowthrough? Typically both are measured (at least on Tara).

From AMT 26 onwards we also measured both temperatures, whereas previously it was just at the intake. In the data files we provide the intake temperature as this is the best representation of the external temperate. We have made this point clear in the relevant sections of the text.

L142 One problem going far into the NIR is the sensitivity to water temperature and its uncertainties.

We have adjusted the text as follows:

*The reference wavelength values are chosen to extend as far as possible into the NIR, where $ap(\lambda)$ becomes increasing negligible with wavelength (Zaneveld and Pegau, 2003). The use of a higher reference ACS wavelength is limited by the presence of spectral anomaly at 740 nm associated with pure water absorption and its high temperature sensitivity (Slade et al., 2010).*

L156 cp is a function of acceptance angle. Thus, while an IOP, it will have a different value for an instrument that has a different one. So the angle is part of its specification.

We have added:

*We also note that $cp(\lambda)$ is a function of acceptance angle, which varies for different optical instruments, and is 0.93◦ for the ACS and AC9 (Boss et al., 2009)*

L158-173 What your a_p uncertainty does not include is that of using this specific scattering correction. Comparison between it and filter-pad technique or integrating spheres suggest a bias in the blue. Lately we found that a combination of Rudiger and Zaneveld yield good agreement in the red AND blue wavelengths with filter pad technique (in an integrating sphere, e.g. Stramski's method) for Tara Europa.

*We agree that we better need to discuss the limitations of the uncertainty propagation (in particular - with respect to the scattering correction), so have added the additional paragraph:*

*Our propagated uncertainties hold for the specific equations used in the scattering correction developed by Zaneveld and Pegau (2003), which assumes negligible ap(λ) in the NIR. This scattering correction is generally applicable to the open ocean and therefore to the dataset in this study. In coastal waters an alternative scattering correction is often used, which allows for non-zero ap(λ) in the NIR (Röttgers et al., 2013). This scattering correction has its own set of measurement equations and therefore would require its own uncertainty propagation scheme to be developed. In future validation of scattering corrections and associated uncertainty in ap(λ), it is recommended to compare with filter-pad estimates. Additionally, following recent developments in above-water radiometry, detailed instrument characterisations (Vabson et al., 2024) and the impact on the uncertainty budget (Lin et al., 2022) should be considered in the analysis.*

L181-183 We find that bubbles typically have little effect on line-height. Could it be Trichodesmium cells or other long chains that may have caused it. Bubbles can be diagnosed, before binning, based on large jump across the two half spectra of the AC-S.

*To be more encompassing, we have re-phrased as: `presence of either bubbles, large particles, or large colonies of phytoplankton cells such as Trichodesmium'.*

*We note that our current QC method used is agnostic about the origin of the spikes. However – for future data collection we will consider the bubble QC diagnostic above.*

L 185 We are finding that we can use a CDOM fluorometer to better interpolate between filter events.

*This is an interesting point for the future, but a CDOM fluorometer was not present on AMT. We therefore have left the text as it is as this method is not applicable to our dataset. We also understand that the CDOM work described above is yet to be published, so cannot be cited.*

L221 There are studies, I am thinking of those by Bricaud, suggesting C* should vary with chlorophyll due to packaging.

*We did (obliquely) discuss this later in the paper in the context of the power-law fitting of parameters A and B  - see eq. (11). We do agree that this point needs to be developed further, so we have added the following text to that section:*

*The HPLC calibration implicitly assumes that there is a linear relationship between Ca(HPLC) and C∗a (ap), or aph(676),in equation (7). In general, the relationship is thought to be non-linear due to the packaging effect (i.e. due to the chlorophyll pigments being packaged within phytoplankton cells, which light absorption levels relative to pigment in solution),
particularly at higher concentrations (Bricaud et al., 1995). However, past analyses of the relationship between C∗a (ap) and Ca(HPLC) in the open ocean (Graban et al., 2020) indicate that the relationship is statistically consistent with being linear for concentration ranges comparable to this study.*

L227 We un-smooth the AC-S spectra as in Chase et al., 2013 prior to computing the line height.

To clarify; no spectral un-smoothing was applied prior to our line-height estimates. This comment relates to Reviewer 2s' point on spectral resolution and bandwidth, so see our response and added text there.

L242 Should it not be in log-space given [chl] distribution in the ocean?

The use of the linear residual metric/calibration approach follows Graban et al. 2020, which we already reference at the top of the section. As we showed that there is a linear (or near-linear) relationship for power-law exponent, B, we can justify the linearity assumption. We also already note that linear percentage residuals are proportional to absolute log differences (see eq. (9)), which provides a way to relate to log-space quantities.

Are you missing a log on the left hand side?

We don't think so, but we can now see that more clarification is needed how the result is obtained, and we have added:

*For a small percentage difference ([C∗a (ap)i]/[Ca(HP LC)i] ≈ 1), it can be shown from a Taylor series expansion of log_e[C∗a (ap)i)/Ca(HP LC)i] about 1 that...*

Fig 2 There seem to be more points below the line. How come?

The plot shows the 1:1 line (rather than the least-squares regression line). We think this should hopefully be clear from the figure legend, so have left it as it is.

Doesn't this indicate one could come with a better global fit than equation (7)?

We do agree that the delta values imply that a better fit could have been obtained if the 0.014 constant (chlorophyll-specific absorption) was left to be a DOF in eq. (7). The approach taken in this paper was to make an initial estimate (referred to as `nominal' in the text) and then make a correction to derive the adjusted Chl value. This was done for consistency with prior work done on the AMT cruises (Graban et al 2020).

We have added the qualifying sentence: *In principle, the chlorophyll-specific absorption could have been set as a degree of freedom in Eq. (7), which would have enabled closer initial fits between HPLC and IOP-derived Tot_Chl_a in Table 4. The approach taken in this paper, where a nominal line-height*

*Tot_Chl_a estimate is made and then corrected, allows us to compare directly with the analysis in Graban et al. (2020) (see their Table 1).*

Not all diatoms are microplankton and diatom are not the only ones with Fuco. See Chase et al., 2020.

*We have rephrased as:*

*Picophytoplankton dominate at lower chlorophyll concentrations, nanophytoplankton at intermediate chlorophyll, and microplankton at higher chlorophyll (Brewin at al. 2019).*

Fig 6 They all look too similar. Typically, less light makes it through to the a-detector than the c-detctor, (e.g. when you do CDOM on both you get more noise in one) so I would have expected differences here.

*We already note in the text that the % uncertainties are much higher for ap than cp (which we believe is in better correspondence with common intuition about `relative noise levels' than the absolute uncertainties).*

*Additionally, to the best of our knowledge, our study is the first to quantify propagated uncertainties (rather than binned variability) for ap and cp.  Fully investigating the uncertainty budget is desirable for future work, but beyond the scope of the current MS. This point was already discussed in the context of scattering correction, along with recommendations for future work in this regard.*

L350: can you say what it is based on, e.g. microscopy or pigments?

Based on pigments - this has been added.

Fig10 Not sure why you need to display both r and r^2 as one is simply the square of the other...

We did discuss the rationale for this in the text. We think r^2 justifies inclusion as it represents the proportion of the variation in the dependent variable that is predictable from the independent variable. Due to quadratic scaling, stronger/weaker correlations are more clearly highlighted in the r^2 plot and are also used in the text discussion to identify thresholds.

L416 This is because the dynamic range of biomass is taken out.

Thanks – we have added `*due the dynamic range of phytoplankton biomass being removed'.*

L443 You have to qualify these by the fact that most cruises happen at the same season. During other seasons these may change.

This was already done at the end of the section, (in the context of differences with other studies). `*The subtle differences between our study and theirs, likely reflects the conditions in the Atlantic*

*Ocean during boreal autumn and austral spring compared to snapshots of more variable conditions from a geographically global database (Boss et al., 2013).'*

Fig 12 Define <a_p> in caption.

Done

L451 It would have been interesting to see if you recover the same correlation between the groups and Gaussian amplitude as in that study, to independently confirm the approach. The code is online in case you are interested.

We agree that this would be an interesting future investigation. We feel that this is beyond the scope of the data release paper in ESSD, and best left for an algorithm-focused follow-on paper, where the methods in Chase *et al.* 2013 are compared alongside other ap-methods of pigment extraction and phytoplankton community structure. We hope that this section will provide some motivation for follow-on work with the dataset.

**Review 2**

Is: "ACS systems (spectral range ~ 400–750 nm at ~ 15 nm spectral resolution)..."

Actually the spectral resolution of the acs instrument is ca. 4.3 nm. According to technical specification provided by manufacturer acs measures absorption and attenuation in the spectral range 400–750 nm at 80 spectral channels which makes 4.3 nm spectral spacing between channels. Please correct.

We previously equated the spectral bandwidth with the resolution rather than the sample spacing, which we agree is ~ 4 nm.  To rectify in the text, we have kept with the terms `spectral sample spacing or `bandwidth' in different contexts.

We now phrase as:

`*(spectral range ~ 400–750 nm at ~ 4 nm spectral sample spacing*).

We have then given more clarification about the bandwidth and how it relates to spectral smoothing (se*e related comment from R1).*

 `*Due to an inbuilt filter, the ACS automatically averages data across its' spectral bandwidth (14-18 nm), which leads to a smoothing of the spectra (Sullivan et al. 2006). Some studies (e.g. Chase et al, 2013) have mitigated for this inbuilt-smoothing using convolution-based methods, but this is not done in this study.'*

*'resolution' was also replaced with 'bandwidth' in the section on ACS/AC9 line height adjustment.*

Page 7, Equation 4

Is: $b'p(\lambda) = a_{p,m}(\lambda) - c_{p,m}(\lambda),$

Shall be

$$b'_{\mathrm{p}}(\mathrm{l}) = c_{\mathrm{p,m}}(\mathrm{l}) - a_{\mathrm{p,m}}(\mathrm{l})$$

Use of Equation 4 as written in the original manuscript would results in negative values of spectral scattering coefficient, as spectral attenuation coefficient values are greater than spectral absorption coefficient values. Please correct.

Thanks for noticing this mistake – we have corrected as suggested.

**Additional changes**

1. Following the editorial feedback, we did colour blindness checks on our images. Figures, 1, 6 and 12 were then replaced with different colour schemes. The other figures all used python colour maps which we believe are colour-blindness compliant.
2. We changed `global median' to `AMT median' in figure 12 and discussion, which better describes the field we are plotting.
3. Uses of `nanoplankton', `microplankton', and `picoplankton' were changed to `nanophytoplankton', `microphytoplankton', and `picophytoplankton' (we noted we were often inconsistent)